# PedVLM: Pedestrian Vision Language Model for Intentions Prediction

## Abstract

Effective modeling of human behavior is crucial for the safe and reliable coexistence of humans and autonomous vehicles. Traditional deep learning methods have limitations in capturing the complexities of pedestrian behavior, often relying on simplistic representations or indirect inference from visual cues, which hinders their explainability. To address this gap, we introduce **PedVLM**, a vision-language model that leverages multiple modalities (RGB images, optical flow, and text) to predict pedestrian intentions and also provide explainability for pedestrian behavior. PedVLM comprises a CLIP-based vision encoder and a text-to-text transfer transformer (T5) language model, which together extract and combine visual and text embeddings to predict pedestrian actions and enhance explainability. Furthermore, to complement our PedVLM model and further facilitate research, we also publicly release the corresponding dataset, PedPrompt, which includes the prompts in the Question-Answer (QA) template for pedestrian intention prediction. PedVLM is evaluated on PedPrompt, JAAD, and PIE datasets demonstrates its efficacy compared to state-of-the-art methods. The dataset and code will be made available at `https://github.com/abc/ped_VLM`.

## 1 Introduction

Understanding human social behavior is crucial for safely deploying autonomous vehicles in an urban environment. In literature, modeling and learning human social behavior is categorized into pedestrian trajectory and intention prediction (Kothari et al., 2021a;b; Liu et al., 2021). Although the former approach has shown effective results in forecasting pedestrian motion using past trajectories, it is often prone to failure when there is an abrupt change in pedestrian dynamics (Kothari et al., 2021a; Sharma et al., 2022). Pedestrian intention prediction[1], on the other hand, offers a more robust approach by anticipating pedestrian decisions before they occur (Hoy et al., 2018). For instance, an intention prediction system can foresee a pedestrian's decision to cross a street well in advance, allowing an autonomous vehicle to adjust its course preemptively. However, predicting pedestrian intentions is complex and requires a holistic approach that integrates context, scene understanding, pedestrian attributes, and careful analysis of past actions.

In the literature, addressing pedestrian intention typically involves using input features such as pedestrian trajectories (Saleh et al., 2019; Bouhsain et al., 2020), environmental context (Rasouli et al., 2017), and social interactions (Helbing & Molnar, 1995; Evans & Norman, 2003). These features are then processed by sequential models, like RNNs (Lorenzo et al., 2020), LSTMs (Zhang et al., 2020), and transformer-based models (Sui et al., 2021), or non-sequential models, such as CNNs and GNN-based models, to improve prediction accuracy and comprehensively understand pedestrian behavior(Saleh et al., 2019; Huang et al., 2021). Although these traditional deep learning-based methods have shown promising results, they often struggle with explainability, particularly in complex scenarios where extracting driving-related knowledge and performing effective reasoning are crucial.

To address these challenges and develop a more holistic model to predict pedestrian intentions, the integration of multiple sources of information, such as vision and text, improves model's explainability and provides a richer contextual representation. Vision-language models (VLMs) have been

---

[1]We use "pedestrian intention" and "pedestrian intent" interchangeably throughout this paper.

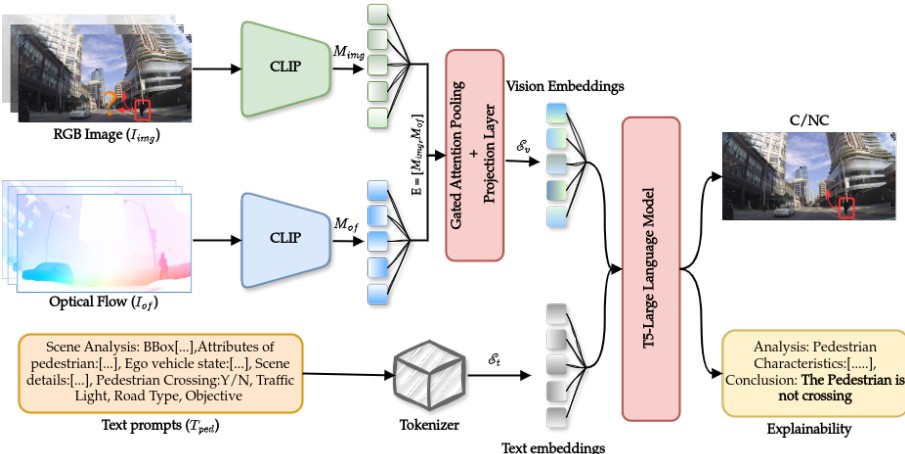

Figure 1: **Overview of PedVLM Framework:** The PedVLM framework consists of two main components: a vision encoder and a large language model. Initially, RGB images and optical flow data are processed through a CLIP-ViT-based vision encoder to create vision embeddings. These embeddings are then combined with tokenized text prompts in a T5-Large Language Model to analyze pedestrian characteristics and scene information. The combined embeddings enable the prediction of pedestrian intentions (crossing/not crossing) and provide insights into the reasoning behind these predictions.

widely adopted for this purpose, leveraging both vision and text modalities to extract comprehensive environmental models (Cui et al., 2024). Recent advances in VLMs have found significant applications in the autonomous driving domain, particularly in understanding driving scenes and informing decision-making processes (Xu et al., 2024). One of the initial works using VLMs for pedestrian intention prediction employed GPT-4V to interpret pedestrian behavior (Huang et al., 2023). This study utilized GPT-4V for zero-shot evaluation on widely adopted pedestrian intention datasets. Although GPT-4V's zero shot evaluation offers insight into how VLMs enhance environmental understanding, especially human-vehicle interactions, the lower results it achieves in comparison to state-of-the-art models show that this approach does not fully exploit the potential of VLMs for task-specific applications.

Building on the insights gained from the limitations of the GPT-4V approach (Huang et al., 2023), in this work, we develop **PedVLM**: **Ped**estrian **V**ision Language **M**odel for Intentions Prediction. PedVLM, as illustrated in Figure 1, employs multiple modalities, including RGB images, optical flow, and text, to extract meaningful information for explaining pedestrian intention in driving scenes. Formally, PedVLM consists of two major components: the vision encoder and the language model. PedVLM utilizes a vision transformer (ViT) variant of CLIP (Radford et al., 2021) for the vision encoder to extract visual embeddings from both RGB images and optical flow data. These embeddings are concatenated and fed into a gated-attention pooling layer to create a unified representation. This visual embdding is then combined with text embeddings and passed to a language model to predict pedestrian actions, such as crossing or not crossing (C/NC), enhancing model explainability. We chose the text-to-text transfer transformer (T5) (Raffel et al., 2020) as the language model due to its low computational and inference cost. A key contribution of our work is creating the PedPrompt dataset, which includes Question-Answer (QA) prompts for pedestrian intention prediction and explainability. To build this dataset, we have employed the TRANS dataset (Guo et al., 2022), integrating publicly available pedestrian intention datasets: JAAD (Rasouli et al., 2017), PIE (Rasouli et al., 2019), and TITAN (Malla et al., 2020). We have evaluated PedVLM's performance on our PedPrompt dataset against various baselines, demonstrating superior efficacy in pedestrian intention prediction-specific metrics and linguistic evaluation metrics. Additionally, we compare PedVLM with state-of-the-art methods on the JAAD and PIE datasets. PedVLM outperforms GPT-4V by 44% in the F1-score and 32.6% in the AUC score on the JAAD dataset. It also shows comparable performance on the PIE dataset against traditional deep-learning methods, as there are no existing evaluation results using VLMs for PIE in pedestrian intention prediction.

The main contributions of our work are:

1. We propose an application specific novel framework, PedVLM, which integrates the sensory inputs with common sense knowledge embedded in the language model to create a contextual representation of the environment and enable joint prediction of pedestrian intentions and the provision of interpretable explanations.

2. Another key contribution of our work is the creation of PedPrompt, a novel dataset specifically designed for pedestrian intention prediction, which includes a comprehensive set of prompts to facilitate research in this area. We make PedPrompt publicly available to support future research and development in the field.

## 2 RELATED WORK

### 2.1 PEDESTRIAN INTENT PREDICTION

Pedestrian intent prediction is an important task that enables safe interactions between automated vehicles and pedestrians. The models for predicting pedestrian intent use as an input various features such as pedestrian bounding boxes (Bouhsain et al., 2020; Yang et al., 2022; Osman et al., 2023; Kotseruba et al., 2021), poses (Fang & López, 2018; Cadena et al., 2019; Lorenzo et al., 2020; Yang et al., 2022; Osman et al., 2023), ego vehicle speed or attributes (Neogi et al., 2020; Kotseruba et al., 2021), visual features (Yang et al., 2022; Sakhai et al., 2024). Many works use combination of different features (Neogi et al., 2020; Piccoli et al., 2020; Yang et al., 2022; Azarmi et al., 2023; Huang et al., 2023; Osman et al., 2023; Munir & Kucner, 2024; Azarmi et al., 2024b;a). In this work, we use visual and text modalities, where the visual features include the RGB images from the scene and optical flow images, and the text prompts to the LLM include information about the pedestrians, the ego vehicle and the environment. Various neural network architectures have been used, such as CNN (Kotseruba et al., 2021; Yang et al., 2022; Azarmi et al., 2023), Graph CNN (Zhang et al., 2022; Cadena et al., 2019), GRU+Attention (Gesnouin et al., 2021; Yang et al., 2022) and Transformers (Lorenzo et al., 2021; Zhou et al., 2023; Osman et al., 2023; Rasouli & Kotseruba, 2023).

### 2.2 LARGE (VISUAL-)LANGUAGE MODELS FOR PEDESTRIAN INTENT PREDICTION

Large language models (LLMs) and large visual-language models (VLMs) have recently been used in the context of predicting pedestrian behavior in driving situations. Recent work (Park et al., 2024) focuses on detecting the pedestrians in the scene, by passing the images and descriptions of appearances of pedestrians to a VLM. A combination of knowledge graphs and LLMs has been used for predicting pedestrian's intention to cross and providing explanation of the decision (Hussien et al., 2024). Zero-shot pedestrian intent prediction with GPT-4V (Huang et al., 2023) shows that the models need to be fine-tuned to adapt to the task of predicting intent. (Gopalkrishnan et al., 2024) address questions regarding driving scenes, including the intention of pedestrians to cross. Drawing inspiration from previous work, we build upon their architecture by integrating optical flow images to capture the scene's motion dynamics. Instead of applying the model for broad scene understanding, we specifically fine-tune it for pedestrian intent prediction, thus enhancing its suitability and performance for this particular task.

## 3 PEDPROMPT DATASET

This section introduces "PedPrompt", our proposed dataset built upon the TRANS dataset (Guo et al., 2022), designed for pedestrian intention prediction using vision-language models. In the following subsections, we will detail the TRANS dataset's composition, explain our prompt generation approach, and outline the PedPrompt dataset statistics.

### 3.1 TRANS DATASET OVERVIEW

The TRANS dataset is built upon three publicly adopted datasets, JAAD, PIE, and TITAN, used for analyzing the pedestrian *"stop"* and *"go"* movements. Since the existing JAAD, PIE, and TITAN

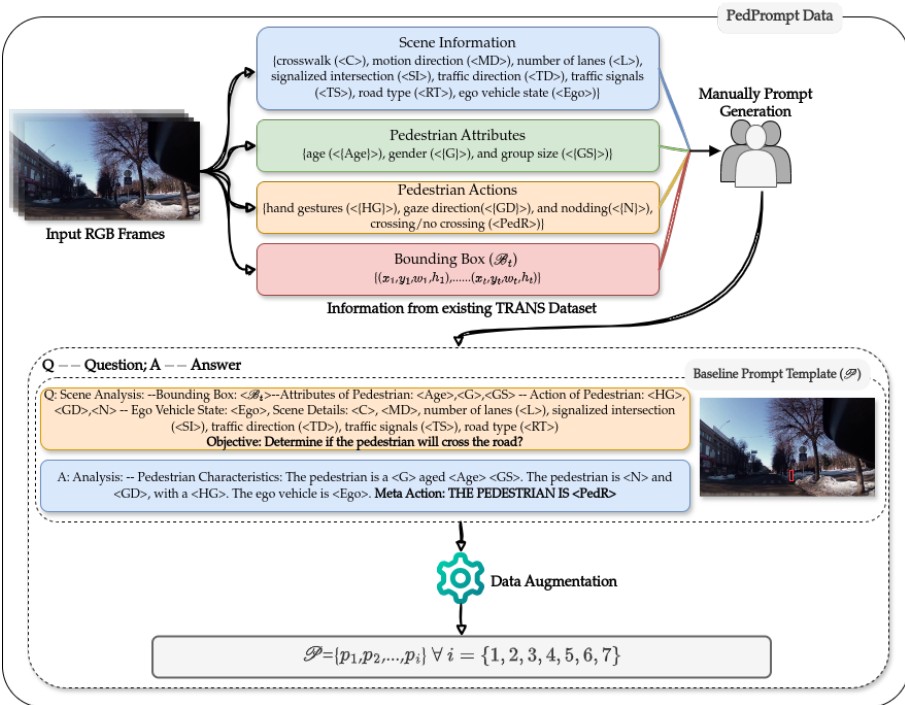

Figure 2: **Overview of PedPrompt Generation:** The generation pipeline involves extracting scene details, pedestrian attributes, actions, and bounding box information from the TRANS dataset. These elements are used to manually create prompts with specific instantiation parameters. A baseline prompt is then formulated in a question-answer template. To enhance linguistic diversity, a data augmentation process generates varied and enriched prompts for the driving scene.

datasets do not provide the annotations for explicitly studying the *"stop"* and *"go"* movements, this limitation is catered by the TRANS dataset. The TRANS dataset is equipped with RGB videos captured from uncalibrated monocular cameras on moving platforms, along with detailed pedestrian localization and walking annotations. In the TRANS dataset, walking motion annotations are integrated with the original datasets, identifying state changes during transition periods. A pedestrian is annotated as *"go"* when transitioning from standing to walking and as *"stop"* for the reverse. To ensure meaningful samples, transitions are considered valid if they last at least $0.5$ seconds, making the dataset more challenging by focusing on pedestrian intentions at critical moments. The TRANS dataset categorizes pedestrians into *"walk"*, *"stand"*, *"stop"*, and *"go"*. To generate the PedPrompt dataset, we classify *"walk"* and *"go"* as crossing actions, while *"stand"* and *"stop"* are considered non-crossing. It is important to note that in the original JAAD, PIE, and TITAN datasets, there is an imbalance in the number of crossing and non-crossing samples, leading to inefficiencies in model learning for pedestrian intention prediction.

## 3.2 Prompt Generation

Transforming raw data into linguistic prompts improves the explainability of pedestrian intentions. To achieve this, we first convert the raw data from the TRANS dataset into text prompts. In generating the PedPrompt dataset, we focus on formulating driving scenes, pedestrian attributes, and actions into QA prompts that clearly articulate pedestrian intentions. A detailed example of prompt generation is illustrated in Figure 2.

To initiate our prompt question generation process, we approach scene analysis from two perspectives: scene information and pedestrian movement within the scene. Formally, to represent pedestrian movement, we extract bounding box information ($\mathcal{B}_t$) from past observations over a specified horizon ($\mathcal{T}_p$), represented as ($\mathcal{B}_t = (x_t, y_t, w_t, h_t)|t = 1, 2, ..., \mathcal{T}_p$), where $(x_t, y_t)$ represent the center coordinates, and $w_t$ and $h_t$ represent the width and height of the pedestrian in the $t$-th image

frame, respectively. During the conversion of $\mathcal{B}_t$, all float-type coordinate values are transformed into text strings with integer representation for consistency and clarity in the generated prompts. To transform the scene information into linguistic prompts, we include crosswalk $< C >$, motion direction ($< MD >$), number of lanes ($< L >$), signalized intersection ($< SI >$), traffic direction ($< TD >$), traffic signals ($< TS >$), road type ($< RT >$) and ego vehicle state ($< Ego >$) as instantiation parameters.

Another crucial aspect of our prompt question generation process is modeling pedestrian behavior. We incorporate pedestrian attributes and actions into our prompts. For attributes, we explore age ($< Age >$), gender ($< G >$), and group size ($< GS >$) as instantiation parameters. Similarly, we have employed hand gestures ($< HG >$), gaze direction ($< GD >$), and nodding ($< N >$) parameters to explain the pedestrian actions. In addition to that, in our prompts template generation, we have also included the pedestrian action in terms of crossing and not crossing represented as ($< PedR >$). Finally, we explicitly add the objective in the question prompts template by asking whether a pedestrian will cross the road. This objective guides the overall analysis and informs the decision-making processes of the proposed PedVLM method.

Since our PedPrompt follows a QA template for prompt generation, our approach to devising answer prompts involves briefly analyzing the scene. This is followed by identifying the pedestrian action as a meta-action, indicating whether they are crossing or not. To enhance the diversity of linguistic prompts, we have augmented the initial prompt set by generating various variants. This is achieved by utilizing instantiation parameters for scene information and pedestrian attributes.

### 3.3 PedPrompt Dataset Statistics

PedPrompt includes a total of $48,696$ prompts in QA format. An overview of the PedPrompt for both questions and answers is illustrated through the word cloud, representing the most frequent terms, as illustrated in Figure 3. From the word cloud of the PedPrompt for both question Figure 3(a) and answer Figure 3(b), a large number of words that describe the PedPrompt dataset are "pedestrian" and "crossing," along with the driving scenario information. PedPrompt dataset statistics reveal a balanced distribution between "crossing" and "not crossing" instances, as shown in the Figure 3(c). Additionally, the pie chart from Figure 3(d) highlights the distribution of context clues, with prediction ($32.6\%$), pedestrian actions ($25.3\%$), perception ($24.2\%$), and scene information ($17.8\%$) contributing to the overall analysis. These insights underscore the comprehensive nature of the dataset, capturing diverse aspects of pedestrian behavior and scene dynamics to enhance model training and evaluation.

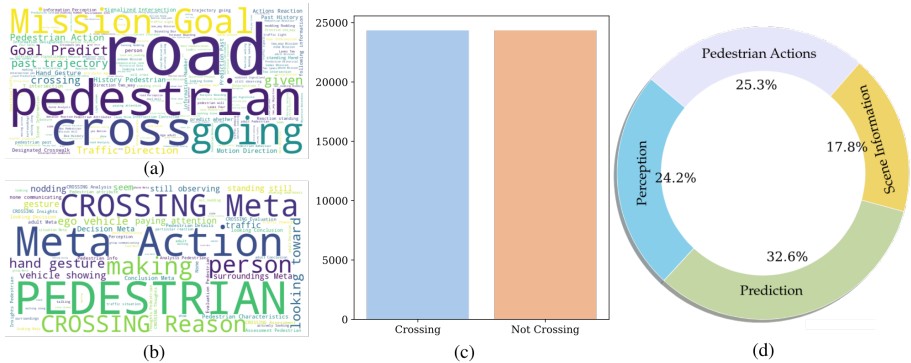

Figure 3: **PedPrompt Data Statistics:** (a) and (b) illustrate the word cloud of most frequent words both in question (Q) and answer (A) respectively. The distrubution of "crossing" and "not crossing" is shown in (c). depicts the distribution of contextual factors influencing pedestrian predictions.

## 4 METHOD

### 4.1 PROBLEM FORMULATION

Given a video sequence $\mathcal{V}$ representing an urban scenario, we define the sequence of observed video frames as $\mathcal{V} = f_1, f_2, \ldots, f_t$, where $t$ represents discrete time steps corresponding to individual image frames $f_t$. Our approach aims to estimate the probability of a pedestrian's intention to cross the street, represented as $I \in [0, 1]$. This prediction leverages multimodal inputs observed over a prior window of $(\mathcal{T}_p)$ time steps, including RGB images ($I_{img}$), optical flow images ($I_{of}$), and text prompts ($T_{ped}$) that encode both past trajectory and contextual information. The text prompts $T_{ped}$ provide the model with the pedestrian's past trajectory, described by a sequence of bounding boxes ($\mathcal{B}_t$). In addition, the text prompts incorporate pedestrian demographic attributes (such as age and gender) and behavioral cues (such as looking, nodding, gesturing, and other non-verbal actions). Scene information—including motion direction, number of lanes, traffic signs, pedestrian crossings, road types, and traffic signals—is also integrated to provide a comprehensive context for accurately predicting the pedestrian's crossing intention.

### 4.2 MODEL ARCHITECTURE

PedVLM as illustrated in Figure 1, encompasses vision and language models for the prediction of pedestrian intentions. Formally, PedVLM's framework is built upon the text-to-text transfer transformer (T5) language model (Raffel et al., 2020), augmented with a vision encoder network to process visual information. The framework integrates RGB images ($I_{img}$) and optical flow ($I_{of}$) as visual modalities, while incorporating textual prompts ($T_{ped}$) derived from contextual features and past pedestrian trajectories. These multimodal inputs are utilized to enhance the model's ability to understand and predict pedestrian behaviors effectively.

To obtain a contextual representation of the driving scene that enables pedestrian intention prediction and provides explainability for those predictions, the embeddings from the vision encoder and language model must be combined. In this study, we adopt the CLIP encoder (Radford et al., 2021) to obtain visual representation. While several versions of the CLIP architecture have been proposed in the literature, we focus on the variant that incorporates a Vision Transformer (ViT) (Alexey, 2020) as the backbone (Gandelsman et al., 2023).

A CLIP encoder is applied to the input RGB image $I_{img} \in \mathbb{R}^{H \times W \times 3}$ to obtain $d$-dimensional representation, denoted as $\text{CLIP}(I_{img})$. The CLIP image representation is obtained by linearly projecting this output into a $d'$-dimensional latent space in the joint vision-and-language space. Formally, let $P_i \in \mathbb{R}^{d' \times d}$ be the projection matrix. The CLIP image embedding is given by, $M_{img} = P_i \cdot \text{CLIP}(I_{img})$. Similarly, the optical flow input $I_{of} \in \mathbb{R}^{H \times W \times 1}$ is processed using the same CLIP architecture, yielding the optical flow embedding, $M_{of} = P_{of} \cdot \text{CLIP}(I_{of})$. The vision embedding can be represented by $E = [M_{img}, M_{of}]$. The parameters of both the CLIP and the projection matrix $P$ are learned during training. Given the individual vision embedding $E_i$, gated pooling attention is used as described in (Wu et al., 2024), which learns a single vision embedding as:

$$\mathcal{E}_v = \sum_{i=1}^{N} \beta_i E_i \qquad (1)$$

Here, $\beta_i$ are the weight of $i$th embedding such that $\sum_{i=1}^{N} \beta_i = 1$, which calculated by:

$$\beta_i = \frac{\exp\left(\alpha^T \left(\tanh\left(ZE_i^T\right) \otimes \text{sigm}\left(HE_i^T\right)\right)\right)}{\sum_{j=1}^{N} \exp\left(\alpha^T \left(\tanh\left(ZE_j^T\right) \otimes \text{sigm}\left(HE_j^T\right)\right)\right)} \qquad (2)$$

Where, $\alpha \in \mathbb{R}^K$ is a weight vector, $Z \in \mathbb{R}^{K \times M}$ and $H \in \mathbb{R}^{K \times M}$ are weight matrices, $M = (d'd)$ is the dimensionality of the embedding and $K$ is a hyperparameter set to 128.

Given the visual embedding ($\mathcal{E}_v$) from gated attention pooling, PedVLM employs a text-to-text transfer transformer (T5) model to integrate visual and linguistic elements seamlessly. The choice of the T5 model is based on two key factors: its lightweight architecture, with fewer than a billion parameters, which reduces computational and inference costs, and its competitive performance across various natural language processing (NLP) tasks (in particular, it has been successfully used

for question-answering in autonomous driving (Gopalkrishnan et al., 2024)). The T5 model utilizes a transformer architecture with an encoder-decoder structure to process input and output text sequences. It employs a stack of transformer layers in both components, incorporating multi-head self-attention and positional encoding to capture hierarchical representations and long-range dependencies. Formally, text prompts ($T_{ped}$) in the QA template are tokenized using the T5 tokenizer, which effectively converts them into text embeddings ($\mathcal{E}_t$). The visual embedding $\mathcal{E}_v \in \mathbb{R}^{d' \times d}$ is projected to match the textual embedding ($\mathcal{E}_t$), allowing them to be concatenated. This concatenation generates enhanced feature maps that contextually integrate information from both modalities. These integrated feature maps are then used to train the T5 language model.

### 4.3 LOSS FUNCTION

In our T5 model, the loss function is based on a standard cross-entropy loss for sequence-to-sequence tasks, but we introduce an additional pedestrian intention loss to better handle the binary classification of pedestrian crossing intentions. Specifically, we observed that language models, when tasked with predicting whether a pedestrian is crossing or not crossing, may struggle to distinguish between these two similar linguistic states, as identified in our initial experiments. To mitigate this issue, we incorporated a pedestrian intention loss, defined using the Binary Cross Entropy (BCE) loss function. This secondary loss improves the model's sensitivity to binary outcomes, particularly the pedestrian's intent to cross or not cross the street. The BCE loss function is calculated as:

$$\mathcal{L}_{\text{int}} = -\frac{1}{N} \sum_{i=1}^{N} (I_i log(\hat{I}_i) + (1 - I_i) log(1 - \hat{I}_i)) \tag{3}$$

where $I$ is the ground truth label (1 for crossing, 0 for not crossing) and $\hat{I}_i$ is the predicted probability. The total loss of our model is defined as:

$$\mathcal{L}_{\text{total}} = (1 - \lambda) \cdot \mathcal{L}_{\text{T5}} + \lambda \cdot \mathcal{L}_{\text{int}}, \tag{4}$$

Where $\mathcal{L}_{\text{T5}}$ is the T5 loss defined in (Raffel et al., 2020), and $\lambda$ weights the balance of the contributions of the T5 loss and the pedestrian intention loss. This dual-loss setup ensures that the model optimizes both the sequence generation and the accuracy of pedestrian intent prediction, crucial for improving decision-making in vision-language tasks.

## 5 EXPERIMENTATION & RESULTS

### 5.1 EXPERIMENTAL SETUP

**Implementation Details:** The proposed PedVLM framework is trained on a GPU server using the PyTorch library, enabling end-to-end training by initializing the network with pre-trained weights from the T5 model and CLIP encoder. The input images are resized to dimensions $[240, 420]$, without additional pre-processing or filtering applied. Optical flow is computed using the PyTorch-based MMFlow toolkit (Contributors, 2021), which incorporates various state-of-the-art techniques. Through extensive experimentation, we selected the Recurrent All Pairs Field Transforms for Optical Flow (RAFT) (Teed & Deng, 2020) method due to its superior performance in capturing detailed motion patterns. The optical flow is computed between consecutive image frames and is resized similarly to dimensions of $[240, 420]$, without further pre-processing. We calculated optical flow for only JAAD (Kotseruba et al., 2016) and PIE (Rasouli et al., 2019) images, TITAN (Malla et al., 2020) already provides the optical flow images. Training optimization is performed using the Adam optimizer, following a learning rate schedule defined as $l_r = l_r^{int} \times (\frac{1-epoch}{max-epoch})^p$, where the initial learning rate $l_r^{int}$ is set to 0.0001. Furthermore, the epsilon and weight decay parameters are configured as $1^{-9}$ and $1^{-4}$, respectively, with the power $p$ set to 0.9 during training. The model is trained for 6 epochs with a batch size of $15$.

**Evaluation Details:** The PedPrompt dataset is utilized to train the PedVLM framework, with the data divided into training, validation, and test sets, following the standard split provided by the TRANS dataset. The model is trained on the training set, with data augmentation applied through variations in text prompts to enhance diversity. Evaluation is conducted on the test set, and additional

evaluations are performed on the JAAD and PIE test datasets without specifically training on their respective training sets. For consistency, we follow the format used in the PedPrompt dataset to generate text prompts for the JAAD and PIE datasets.

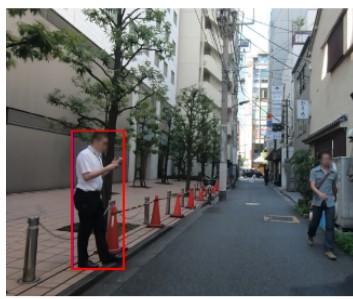

Question: Scene Analysis: —Bounding Box History— [(574.0, 704.0, 823.0, 1380.0], [462.0, 669.0, 754.0, 1456.0], [301.0, 631.0, 665.0, 1520.0], [55.0, 572.0, 491.0, 1520.0], [1.0, 473.0, 214.0, 1520.0]] —Pedestrian Attributes:— Age: adult —Pedestrian Behavior: looking into phone —Pedestrian Action: None, —Objective: Determine if the pedestrian will cross the road

Predicted Answer: Analysis: — Pedestrian Characteristics: The pedestrian is looking into phone and doing nothing and is adult. Conclusion: Meta Action: **THE PEDESTRIAN IS NOT CROSSING.**

Actual Answer: Analysis: — Pedestrian Characteristics: The pedestrian is looking into phone and doing nothing and is adult. Conclusion: Meta Action: **THE PEDESTRIAN IS NOT CROSSING.**

(a)

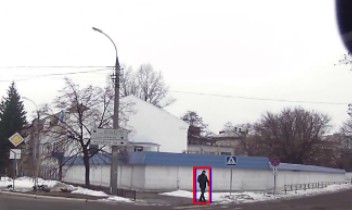

Question: Scene Analysis: —Bounding Box History— ([1043.0. 651.0. 10E13.0, 757.0]. [1012.0. 644.0. 1054.0. 754.0]. [979.0. 644.0. 1021.0. 759.0]. [946.0. 650.0. 989.0. 768.0]. [917.0. 654.0. 961.0. 776.0]] —Pedestrian Behavior—Reaction: clear path, Hand Gesture: No, Look: not nodding, Nodding: no —Environment Info: —Designated Crosswalk: yes —Motion Direction: longitudinal —Number of Lanes Three —Intersection: yes —Signalized Intersection: no —Traffic Dinsotion: one-way. —Traffic Light: unkown. —Goal: Will the pedestrian cross the road?

Predicted answer: Insights — Pedestrian Info: The pedestrian is making no hand gesture and is not nodding and Not-looking toward the ego vehicle. Decision:Meta Action: **THE PEDESTRIAN IS CROSSING.**

Actual answer: Insights — Pedestrian Info: The pedestrian is making no hand gesture and is not nodding and looking toward the ego vehicle. Decision:Meta Action: **THE PEDESTRIAN IS CROSSING.**

(b)

Figure 4: Example from the data: scene image with a pedestrian bounding box; prompt to the model, containing pedestrian and environment information; predicted and expected answer from the model.

## 5.2 RESULTS

In our experimental evaluation, we assess PedVLM's performance in explaining pedestrian interactions from two distinct perspectives. Firstly, we employ pedestrian intention prediction metrics, including F1-score, Area under the curve (AUC), precision, and recall, to evaluate the model's accuracy in predicting pedestrian behavior. Secondly, we utilize linguistic evaluation metrics to assess the quality of the generated explanations. These metrics include BLEU-4, which measures the overlap of 4-grams between the generated and reference texts (Papineni et al., 2002); METEOR, which evaluates the alignment between the output and reference texts (Banerjee & Lavie, 2005); ROUGE-L, which determines sentence similarity by identifying the longest common subsequence (Lin, 2004); and CIDEr, which accounts for lexical and semantic similarity between the generated and reference texts (Vedantam et al., 2015). This comprehensive evaluation approach allows us to assess both the predictive accuracy of PedVLM and the linguistic quality of its generated explanations, providing a holistic view of the model's performance in interpreting and describing pedestrian interactions.

In the baseline methods, we employ different variants of image encoder, to quantitatively assess the vision encoder's effectiveness. We experiment with CLIP, ViT, and ResNet50 as a vision encoder network for both RGB image and optical flow data to extract the vision embeddings ($\mathcal{E}_v$). Additionally, we experiment with two variants of the T5 language model: T5-base and T5-large. Table 1 illustrates the quantitative results of the baseline method on pedestrian intention evaluation metrics, including F1-score, AUC, precision, and recall. Similarly, the performance of baseline methods on the linguistic evaluation is also shown in Table 1. From our experimental results, the PedVLM baseline method with T5-base and CLIP as vision encoder highlights better performance in both pedestrian intention-specific and also on linguistic evaluation metrics. Specifically, T5-base-CLIP baseline method achieves an F1-score of 0.6733, an AUC score of 0.6586, a precision score of 0.6450, and a recall of 0.7035. Although the T5-base-CLIP method has a lower precision score in contrast to the T5-large-ViT baseline, the most critical metrics for pedestrian intention prediction

Table 1: Performance Comparison of Models in Intention Prediction and Linguistic Evaluation

| Models | Vision Encoder | Intention Prediction Evaluation | | | | Linguistic Evaluation | | | |
|--------|----------------|------|------|-----------|--------|--------|--------|---------|-------|
| | | F1 | AUC | Precision | Recall | BLEU-4 | METEOR | ROUGE-L | CIDEr |
| T5-Base | ResNet50 | 0.6101 | 0.6290 | 0.6432 | 0.5800 | 91.12 | 64.10 | 92.30 | 9.727 |
| T5-Base | ViT | 0.6281 | 0.5806 | 0.5642 | 0.5642 | 92.42 | 63.54 | 94.15 | 9.74 |
| T5-Base | CLIP | **0.6733** | **0.6586** | 0.6450 | **0.7035** | **98.30** | **71.70** | **99.40** | **9.80** |
| T5-Large | ResNet50 | 0.6448 | 0.6196 | 0.6047 | 0.6906 | 85.40 | 58.62 | 81.30 | 8.57 |
| T5-Large | ViT | 0.4662 | 0.6020 | **0.7089** | 0.3473 | 86.63 | 60.01 | 87.31 | 9.59 |
| T5-Large | CLIP | 0.6006 | 0.6236 | 0.6396 | 0.5660 | 87.97 | 61.31 | 88.23 | 9.63 |

are the F1-score and AUC. In these evaluation metrics, the T5-large-ViT does not outperform the T5-base-CLIP. These experimental findings motivate us to select T5-base-CLIP as the proposed solution for PedVLM. Furthermore, we have also evaluated the baseline methods in terms of linguistic evaluation metrics as illustrated in Table 1. In terms of linguistic evaluation, the only difference between the "crossing" and "not crossing" predictions is the word "not." This minimal difference explains why our baseline achieves higher scores in BLEU-4, METEOR, ROUGE-L, and CIDEr metrics. Specifically, our T5-base-CLIP baseline outperforms the other baselines, achieving scores of 98.30 for BLEU-4, 71.70 for METEOR, 99.40 for ROUGE-L, and 9.80 for CIDEr. In addition to quantitative results, Figure 4 illustrates the qualitative results of T5-base-CLIP on PedPrompt dataset (More qualitative results in Appendix A).

In addition to comparing baseline methods, we evaluate the performance of PedVLM with other state-of-the-art methods. For a fair comparison with the state-of-the-art method, we have opted to use JAAD and PIE datasets, focusing solely on pedestrian intention-specific evaluation metrics. Ped-VLM demonstrates better performance in contrast to state-of-the-art methods on the JAAD dataset, as illustrated in Table 2. Since our PedVLM addresses the problem of pedestrian intention using vision-language models, we compare PedVLM with GPT-4V (Huang et al., 2023), which also employs this approach, but makes a zero-shot prediction. PedVLM outperforms GPT-4V by $44\%$ in the F1-score and $32.6\%$ in the AUC score, and it also achieves higher precision and recall. Additionally, we also compare the PedVLM performance on the JAAD dataset with the other traditional state-of-the-art pedestrian intention prediction methods as detailed in Table 2. We also assess PedVLM's performance using the PIE dataset. To our knowledge, no vision-language model has previously utilized the PIE dataset for evaluation. Therefore, we compare PedVLM with traditional pedestrian intention prediction methods. PedVLM's performance on the PIE dataset is suboptimal compared to other state-of-the-art algorithms. This is attributed to the dataset's characteristics, which include sequences of pedestrians that are often too distant to be clearly visible in the image, as well as frequent occlusions (see Appendix for examples). Conversely, the JAAD dataset does not contain such sequences. This disparity in performance suggests that PedVLM's accuracy is contingent upon contextual features in the images, despite the provision of past trajectory information in the text prompt. Notably, pedestrians at a distance from the ego vehicle pose a challenge for analysis, a limitation that is also applicable to human observers. This highlights the importance of considering contextual factors in the development of pedestrian intention prediction models.

Table 2: Comparison of our method against several baseline models on JAAD and PIE datasets.

| Models | Model variants | Input | | JAAD Dataset | | | | PIE Dataset | | | |
|--------|----------------|-------|-----------|------|------|-----------|--------|------|------|-----------|--------|
| | | Frames | Extra info | F1 | AUC | Precision | Recall | F1 | AUC | Precision | Recall |
| PCPA | 3D CNN+RNN+Attention | 16 | x | 0.71 | 0.5 | - | - | 0.77 | 0.86 | - | - |
| TrouSPI-Ne | GRU+Attention | 16 | x | 0.76 | 0.56 | 0.66 | 0.91 | 0.8 | 0.88 | 0.73 | **0.89** |
| IntFormer | Transformer | 16 | x | 0.69 | 0.54 | - | - | 0.81 | **0.92** | - | - |
| ST CrossingPose | Graph CNN | 16 | x | 0.74 | 0.56 | 0.66 | 0.83 | - | - | - | - |
| FFSTP | GRU+Attention | 16 | Seg | 0.74 | 0.54 | 0.65 | 0.85 | - | - | - | - |
| Pedestrian Graph + | Graph CNN+Attention | 32 | Seg,P3D | 0.76 | 0.7 | 0.77 | 0.75 | 0.81 | 0.9 | 0.83 | 0.79 |
| PIT-Block(a) | Transformer | 16 | x | 0.81 | 0.65 | 0.71 | 0.93 | **0.82** | 0.9 | **0.85** | 0.79 |
| PIT-Block(d) | Transformer | 16 | x | 0.76 | 0.69 | 0.79 | 0.74 | 0.81 | 0.91 | 0.82 | 0.8 |
| GPT-4V | Transformer | 10 | text prompt | 0.65 | 0.61 | 0.82 | 0.54 | - | - | - | - |
| GPT-4V Skip | Transformer | 10 | text prompt | 0.64 | 0.59 | 0.81 | 0.53 | - | - | - | - |
| PedVLM (Ours) | CLIP+ T5-base | 5 | Optical Flow + text prompt | **0.9363** | **0.8094** | **0.8920** | **0.9853** | 0.7358 | 0.51718 | 0.6512 | 0.8457 |

### 5.2.1 ABLATION STUDY

We choose as our basic setup T5-Base model with CLIP vision encoder, $\lambda = 0.5$ and using all data modalities (RGB images, optical flow and text prompts) as input. We explore what are the effects of using different type of visual features and what is the importance of the two losses we implement. In all ablation experiments, we train the models on the PedPrompt training set and show the results on the PedPrompt test set.

**Effect of using different visual features:** We evaluate how each vision modality, RGB images and optical flow contribute for solving the task of pedestrian intent prediction. For this ablation, we use the base setup, i.e. T5-Base, CLIP vision encoder, $\lambda = 0.5$. We conduct an ablation test with the following modalities for the input: (1) All data modalities - RGB images, optical flow and text prompt containing the scene information (`RGB+OF+Text`); (2) RGB images and text prompt, without optical flow (`RGB+Text`); (3) Optical flow and text prompt, without RGB images (`OF+Text`). The results from the comparison in Figure 5(a) show that combining both the RGB images and optical flow are beneficial for predicting the pedestrian intention for crossing.

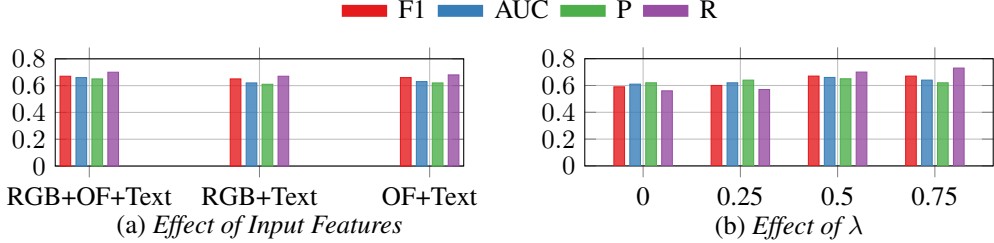

(a) *Effect of Input Features*  (b) *Effect of $\lambda$*

Figure 5: Ablation on the effect of using visual features (left) and $\lambda$ (right), evaluated on the Ped-Prompt test set.

**Effect of $\lambda$ for balancing the two loss functions:** We evaluate the contribution of the secondary loss functions for training on the pedestrian intent prediction task. We experiment with $\lambda \in [0.0, 0.25, 0.5, 0.75, 1]$, where $\lambda$ corresponds to the weight of the intent classification loss function $\mathcal{L}_{\text{int}}$, i.e. for $\lambda = 0$ only the text generation loss is updated. We use the base setup, i.e. T5-Base with CLIP vision encoder, and all data modalities (`RGB+OF+Text`) and we only vary the value of $\lambda$. The results, shown in Figure 5(b), indicate that introducing a task-specific loss function is beneficial, and using equal weight to both losses performs best. However, using only the task-specific loss function and not optimizing the text generation capabilities of the model, fails to correctly predict the intention of crossing.

## 6 CONCLUSION

In this work, we introduce PedVLM, a novel task-specific vision-language model designed to predict pedestrian intentions and provide interpretable explanations for those predictions. PedVLM leverages a CLIP-based vision encoder and a T5-based language model to learn a rich feature representation from multimodal data, including RGB images, optical flow, and text. Furthermore, we release the PedPrompt dataset, a comprehensive collection of QA prompts specifically designed for pedestrian intention prediction. Our experimental results demonstrate that PedVLM outperforms state-of-the-art methods in both predicting pedestrian intentions and providing explanations for those predictions.

While our work represents a significant advancement in the development of vision-language models for pedestrian intention prediction, we believe that there are opportunities for further improvement. Specifically, integrating vision-language models with other modalities, such as lidar and radar, could enhance the accuracy and robustness of pedestrian intention prediction. Additionally, the PedVLM and PedPrompt framework can be extended to other applications, including pedestrian trajectory prediction and human-vehicle interaction analysis, offering a promising direction for future research.

# 7 REPRODUCIBILITY STATEMENT

To ensure the reproducibility of our work, we have taken several measures, which are detailed throughout the paper. The full implementation of our proposed framework, including the Ped-VLM model, data preprocessing, training scripts, and evaluation protocols, will be made available Github. For dataset-related reproducibility, we provide a comprehensive description of the Ped-Prompt dataset and a detailed explanation of how we processed and evaluated the JAAD and PIE datasets. Additionally, all hyperparameters, model configurations, and training details are described in Section 5 to facilitate replication of our results.

# 8 CODE OF ETHICS

In this research, no human subjects or personal data were involved. All experiments were conducted using publicly available datasets. We strictly adhered to ethical guidelines and best practices in machine learning research, ensuring compliance with relevant standards in data privacy, security, and fairness. We prioritized the ethical use of resources and maintained transparency and reproducibility throughout the research process.

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

## A   APPENDICES

### A.1   EXAMPLES FOR PROMPTS

We show examples from the PedPrompt instances, where we have a scene image with a pedestrian marked with a bounding box, as well as the prompt to the model, the expected response and the output from the model as illustrated in Figure 6.

### A.2   FAILURE CASES FROM THE PIE DATASET

Although our model shows good results on JAAD, it performs poorly on the PIE dataset (see Table 2). We examined examples from both datasets and empirically noticed that that PIE contains many examples of predictions for pedestrians who are far from the ego vehicle, are hidden behind obstacles and it is difficult even for a human to figure out their crossing intention (Figure 7, Figure 8). This disparity in performance suggests that PedVLM's accuracy is contingent upon contextual features in the images, despite the provision of past trajectory information in the text prompt. Notably, pedestrians at a distance from the ego vehicle pose a challenge for analysis, a limitation that is also applicable to human observers. This highlights the importance of considering contextual factors in the development of pedestrian intention prediction models

### A.3   EXAMPLES FROM THE JAAD DATASET

From qualitative evaluation, we noticed that the JAAD dataset contains more examples where the initial frame for one pedestrian is clear also for a human - either the pedestrians are close to the ego vehicle, or, if they are far from it, they are usually not hidden by another object. We show several examples where the model correctly predicts "crossing" (Figure 9) or "not crossing" (Figure 10).

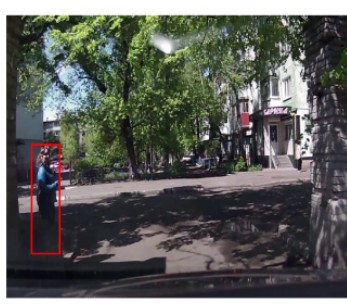

Question: Scene Analysis: —Bounding Box History:— [[181.0, 448.0, 300.0, 794.0], [135.0, 444.0, 267.0, 823.0], [82.0, 440.0, 230.0, 849.0], [38.0, 436.0, 200.0, 871.0], [22.0, 432.0, 182.0, 883.0]] —Actions of Pedestrian:— Reaction: no reaction —Hand Gesture: no —Look: looking —Nodding: not nodding —Scene Information: —Designated Crosswalk: no —Motion Direction: latitudinal —Number of Lanes: One — Intersection: yes —Signalized Intersection: no information —Traffic Dinsclion: two—way. The traffic light is unkown., -Objective: Determine if the pedestrian wit cross the road ?

Predicted Answer: Analysis: — Pedestrian Characteristic.: The pedestrian is making no hand gesture and is not nodding and looking toward the ego vehicle. Conclusion:Meta Action: **THE PEDESTRIAN IS CROSSING.**

Actual Answer: Analysis: — Pedestrian Characteristic.: The pedestrian is making no hand gesture and is not nodding and looking toward the ego vehicle. Conclusion:Meta Action: **THE PEDESTRIAN IS CROSSING.**

(a)

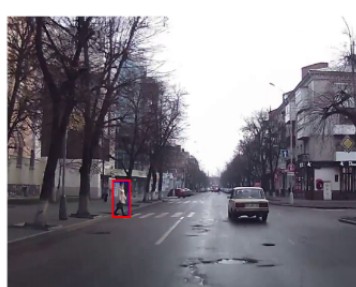

Question: Prediction Task —Bounding Box— [[585.0, 672.0, 629.0, 768.0], [569.0, 674.0, 615.0, 779.0], [547.0, 652.0, 595.0, 769.0], [521.0, 637.0, 566.0, 762.0], [490.0, 656.0, 536.0, 794.0]] —Pedestrian Behavior —Reaction: clear path, Hand Gesture: looking, Look: not nodding, Nodding: no —Environment Info: — Designated Crosswalk: yes —Motion Direction: latitudinal —Number of Lanes Three —Intersection: yes — Signalized Intersection: no —Traffic Dinsotion: two-way. —Traffic Light: unkown. —Goal: Will the pedestrian cross the road?

Predicted answer: Insights — Pedestrian Info: The pedestrian is making no hand gesture and is not nodding and looking toward the ego vehicle. Decision:Meta Action: **THE PEDESTRIAN IS CROSSING.**

Actual Answer: Insights — Pedestrian Info: The pedestrian is making no hand gesture and is not nodding and looking toward the ego vehicle . Decision:Meta Action: **THE PEDESTRIAN IS CROSSING.**

(b)

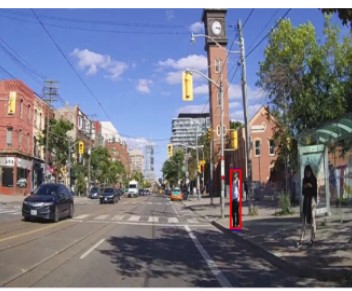

Question: Observation and Forecast: —Bounding Box History— [[1184.97, 706.86, 1218.31, 846.1], [1231.12, 693.82, 1272.37, 856.69], [1298.88, 674.5, 1358.54, 883.26], [1410.04, 637.35, 1479.18, 913.9], [1591.03, 583.41, 1684.26, 966.04]] —Pedestrian Actions:— Reaction: standing, Hand Gesture: looking, Look: not nodding, Nodding: no —Scene Context —Crosswalk no —Motion Direction: no information —Number of Lanes: Four —Intersection: T-intersection —Signalized Intersection: Controlled —Traffic Direction: two_way. ObJective: Predict if the pedestrian will cross the road ?

Predicted Answer: Evaluation: — Pedestrian Details: The pedestrian is making no hand gesture and is not nodding and looking toward the ego vehicle. Conclusion:Meta Action: **THE PEDESTRMN IS NOT CROSSING.**

Actual Answer: Evaluation: — Pedestrian Details: The pedestrian is making no hand gesture and is not nodding and looking toward the ego vehicle . Conclusion:Meta Action: **THE PEDESTRMN IS NOT CROSSING.**

(c)

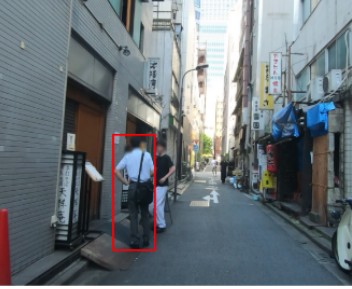

Question: Scene Analysis: —Bounding Box History— [((982.0. 751.0. 1194.0. 1276.01 [924.0. 744.0. 1154.0. 1317.0]. [843.0. 726.0. 1100.0. 1363.0]. [728.0. 708.0 1017.0. 1425.0]. [572.0. 685.0. 908.0. 1498.0]] —Pedestrian Attributes:— Age: adult —Pedestrian Behavior: talking in group —Pedestrian Action: None, —Objective: Determine if the pedestrian will cross the road ?

Predicted Answer: Analysis: — Pedestrian Characteristics: The pedestrian talking in the group and doing nothing and is adult. Conclusion: Meta Action: **THE PEDESTRIAN IS NOT CROSSING.**

Actual Answer: Analysis: — Pedestrian Characteristics: The pedestrian is talking in the group and doing nothing and is adult. Conclusion: Meta Action: **THE PEDESTRIAN IS NOT CROSSING.**

(d)

Figure 6: Examples from the PedPrompt dataset: scenes with pedestrian bounding box, prompt to the LLM, expected response and model output.

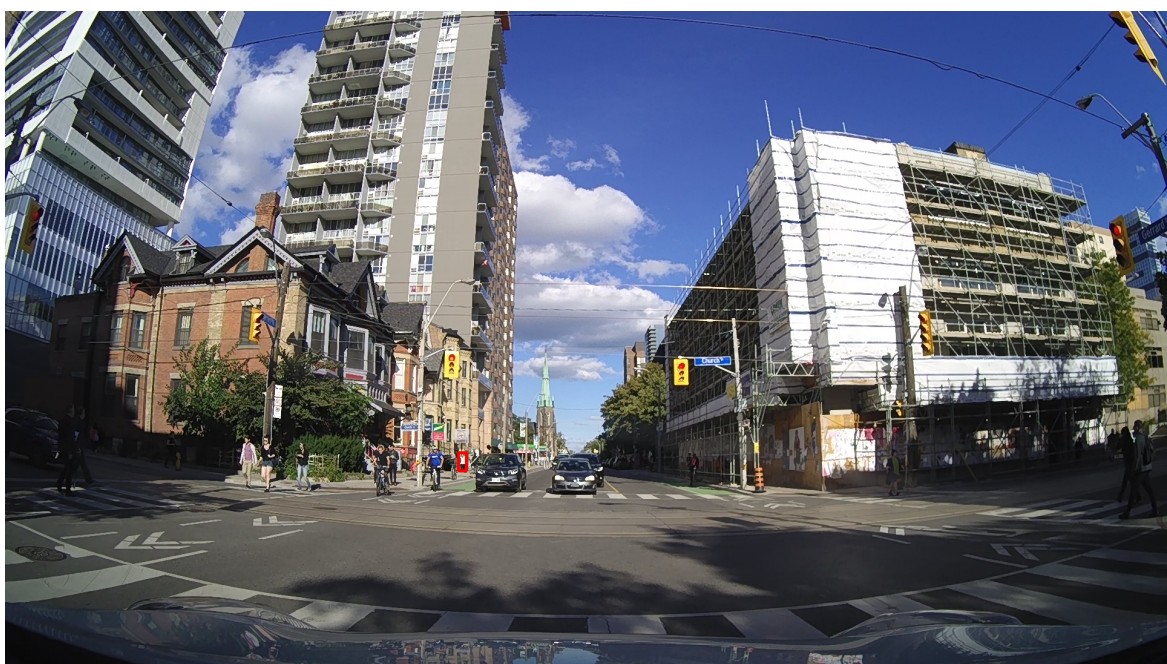

Question: Analysis and Prediction: —Historical Data:— [[749, 729, 13, 30], [749, 729, 13, 30], [749, 729, 13, 30], [749, 729, 13, 30], [748, 729, 13, 31]] —Pedestrian Info:— Age: adult, Gender: female, —Behavior of Pedestrian:— Reaction: walking, Hand Gesture: no, Look: not—looking, —Scene Details: —Intersection: four—way intersection, Lane Count: Four, Signalized Intersection: Controlled and Signalized, Traffic Flow: two_way —Mission: Will the pedestrian cross the road?
Predicted Answer: Assessment: — Pedestrian Characteristics: The pedestrian is making no hand gesture and is not—looking toward the ego vehicle, showing a not—looking. Decision:Meta Action: THE PEDESTRIAN IS CROSSING
Actual Answer: Assessment: — Pedestrian Characteristics: The pedestrian is a female aged adult. They are walking and not—looking towards the ego vehicle, showing a no hand gesture. Decision: Meta Action: THE PEDESTRIAN IS NOT CROSSING.

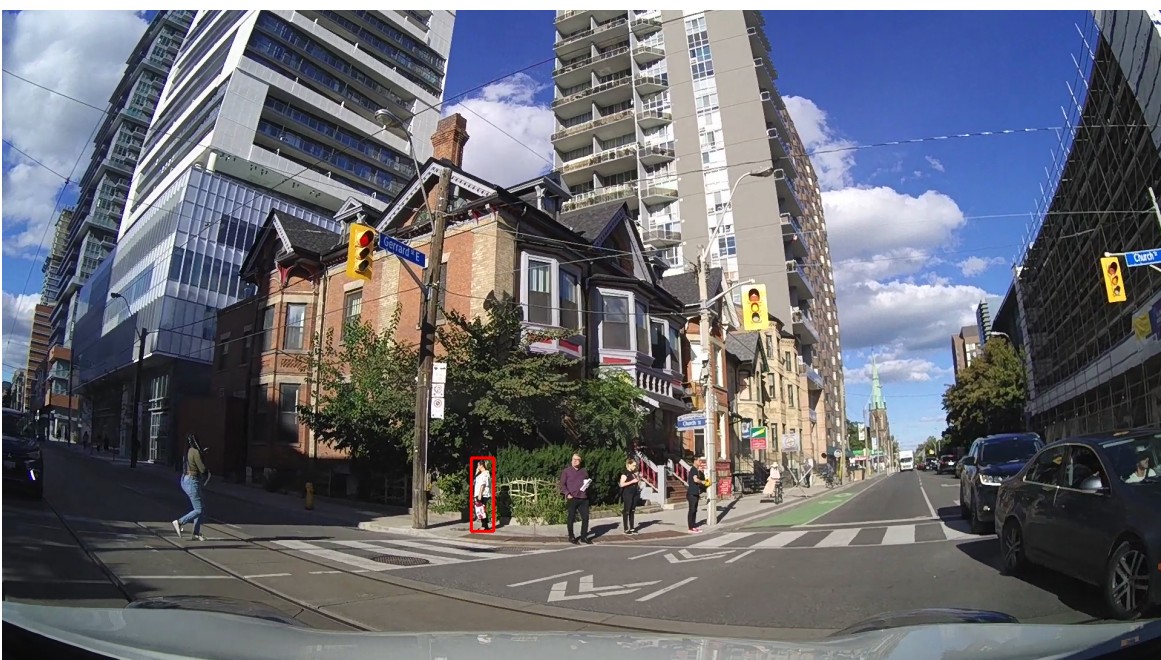

Question: Prediction Task: —Historical Bounding Box:— [[775, 739, 37, 121], [787, 740, 37, 121], [800, 740, 38, 121], [812, 741, 39, 122], [826, 742, 40, 122]] —Pedestrian Details:— Age: adult, Gender: female, —Pedestrian Behavior:— Action: standing, Hand Gesture: no, Look: not—looking, —Environment Info: —Intersection: four—way intersection, Lane Count: Four, Signalized Intersection: Controlled and Signalized, Traffic Flow: two_way—Goal: Will the pedestrian cross the road? Predict their trajectory for the next 5 frames.
Predicted Answer: Insights: — Pedestrian Info: The pedestrian is making no hand gesture and is not—looking toward the ego vehicle, showing a not—looking. Decision:Meta Action: THE PEDESTRIAN IS NOT CROSSING
Actual Answer: Insights: — Pedestrian Info: The pedestrian is a female aged adult . They are standing and not—looking towards the ego vehicle, showing a no hand gesture. Decision: Meta Action: THE PEDESTRIAN IS NOT CROSSING

Figure 7: Failure case from the PIE dataset, where the model incorrectly predicts "crossing" when the pedestrian is hardly visible (top), and the prediction is correct when the pedestrian is close to the ego vehicle (bottom).

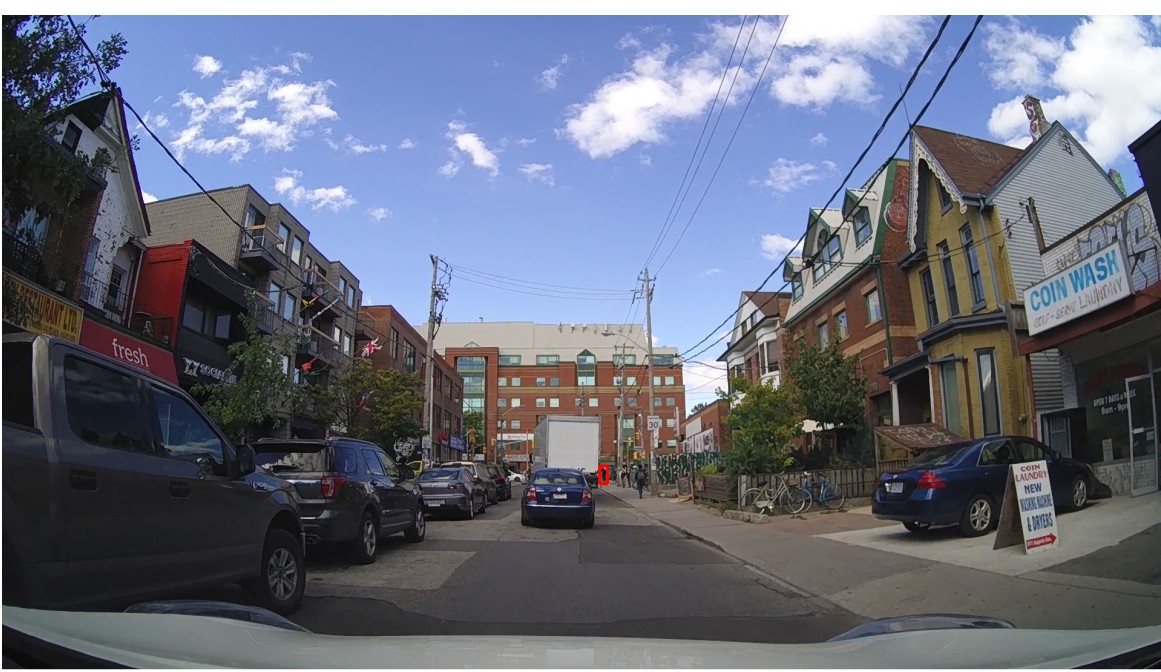

Question: Scene Analysis: —Bounding Box History:— [[987, 744, 13, 29], [987, 744, 13, 29], [987, 744, 13, 29], [987, 743, 13, 29], [988, 743, 13, 29]] —Attributes of Pedestrian:— Age: adult,
Gender: male, —Actions of Pedestrian:— Action: standing, Hand Gesture: no, Look: not—looking, —Scene Details: —Intersection: T—left intersection, Lanes: Four, Signalized Intersection: Controlled and
Signalized, Traffic Direction: two_way   —Objective: Determine if the pedestrian will cross the road ?
Predicted Answer: Analysis: — Pedestrian Characteristics: The pedestrian is making no hand gesture and is not—looking toward the ego vehicle, showing a not—looking. Conclusion:Meta Action: THE PEDESTRIAN IS NOT CROSSING
Actual Answer: Analysis: — Pedestrian Characteristics: The pedestrian is a male aged adult. They are standing and not—looking towards the ego vehicle, with a no hand gesture. Conclusion: Meta Action: THE PEDESTRIAN IS CROSSING

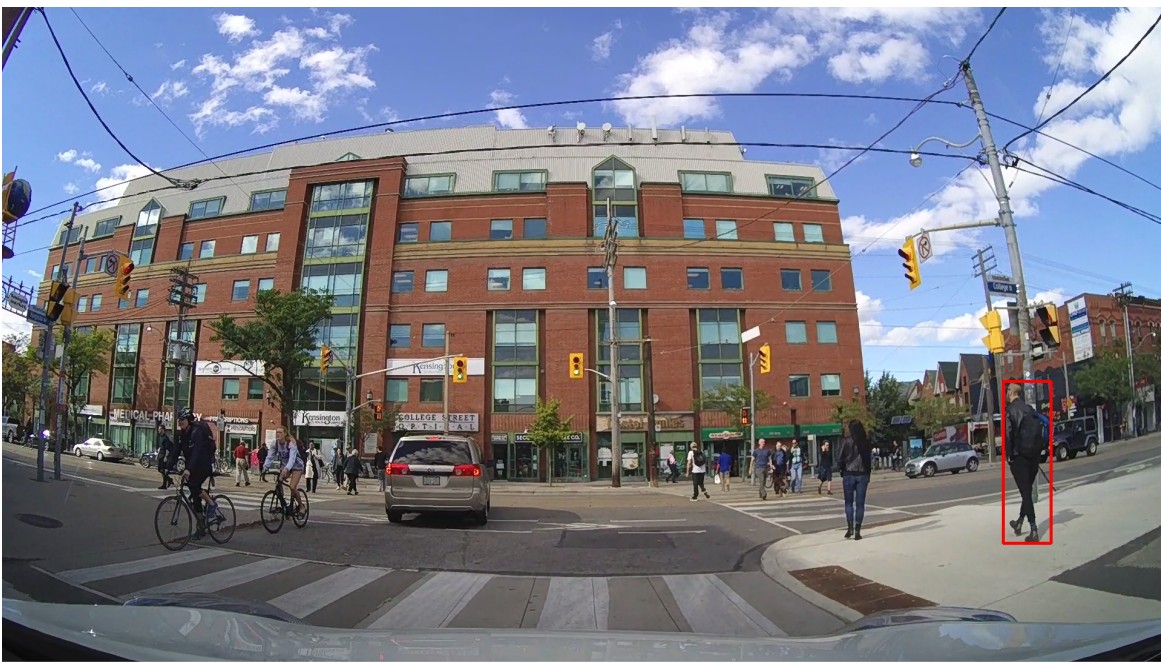

Question: Perception and Prediction: —Past History:— [[1654, 617, 79, 267], [1665, 613, 82, 272], [1676, 609, 85, 276], [1687, 606, 88, 281], [1698, 602, 91, 288]] —Pedestrian Attributes:— Age:
adult —Gender: male   —Pedestrian Actions:— Action: walking —Hand Gesture: no —Look: not—looking —Scene Information: —Intersection: T—left intersection   —Number of Lanes: Four —Signalized
Intersection: Controlled and Signalized —Traffic Direction: two_way   —Mission Goal: Predict if the pedestrian is going to cross the road or not?
Predicted Answer: Thoughts: — Pedestrian attribute from Perception: The pedestrian is making no hand gesture and is not—looking toward the ego vehicle, showing a not—looking. Meta Action: THE PEDESTRIAN IS CROSSING
Actual Answer: Thoughts: — Pedestrian attribute from Perception: The pedestrian is a male adult . The pedestrian is walking and is not—looking toward the ego vehicle, showing a no hand gesture. Meta
Action: THE PEDESTRIAN IS CROSSING.

Figure 8: Failure case from the PIE dataset, where the model incorrectly predicts "not crossing" when the pedestrian is far (top), and the prediction is correct when the pedestrian is close to the ego vehicle (bottom).

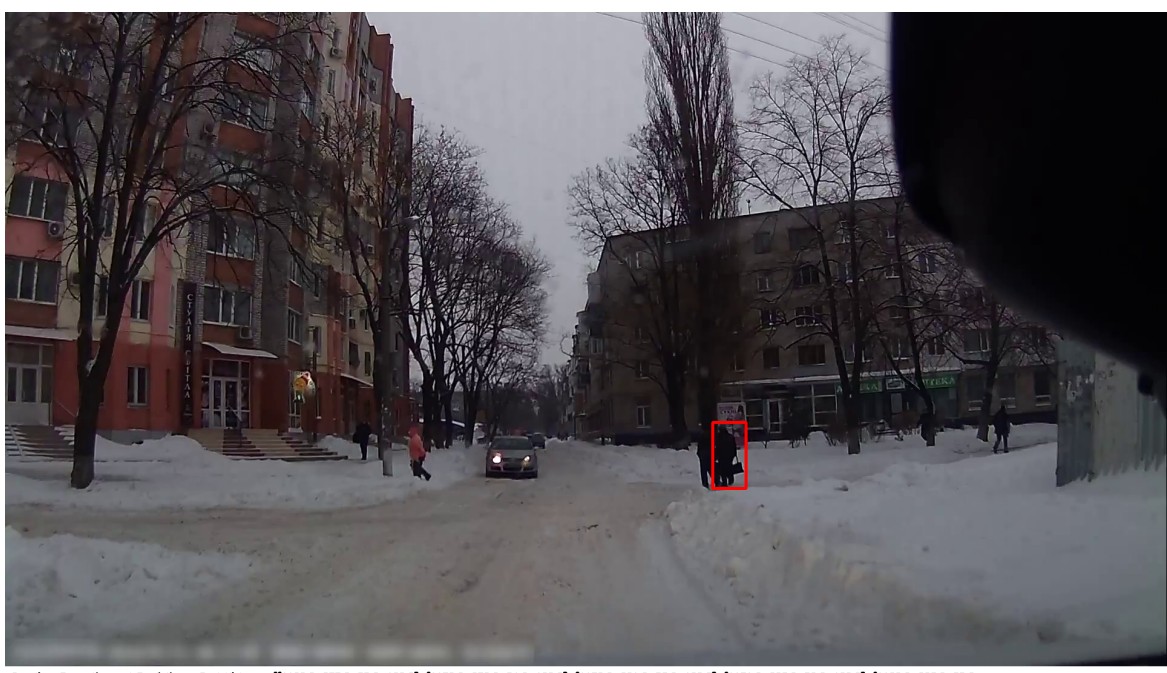

Question: Perception and Prediction: —Past History:— [[1169.0, 677.0, 55.0, 109.0], [1170.0, 679.0, 54.0, 109.0], [1170.0, 681.0, 53.0, 108.0], [1171.0, 683.0, 52.0, 108.0], [1168.0, 683.0, 55.0, 109.0]] —Pedestrian Attributes:— Age: senior —Gender: female —Group Size: group ofTwo —Pedestrian Actions:— Reaction: no reaction —Hand Gesture: no —Look: not—looking —Nodding: not_nodding —The ego vehicle is stopped —Scene Information: —Designated Crosswalk: yes —Motion Direction: latitudinal —Number of Lanes: Two —Signalized Intersection: no —Traffic Direction: two_way —Pedestrian Crossing: no —Pedestrian Sign: no —Stop Sign: no —Traffic Light: no information —Road Type: street —Mission Goal: Predict if the pedestrian is going to cross the road or not?
Predicted Answer: Thoughts: — Pedestrian attribute from Perception: The pedestrian is making no hand gesture and is not_nodding and not—looking toward the ego vehicle, showing a not—looking. Meta Action: THE PEDESTRIAN IS CROSSING.
Actual Answer: Thoughts: — Pedestrian attribute from Perception: The pedestrian is a female senior walking in a group of group ofTwo. The pedestrian is not_nodding and is not—looking toward the ego vehicle, showing a no hand gesture. The ego vehicle is stopped.Meta Action: THE PEDESTRIAN IS CROSSING.

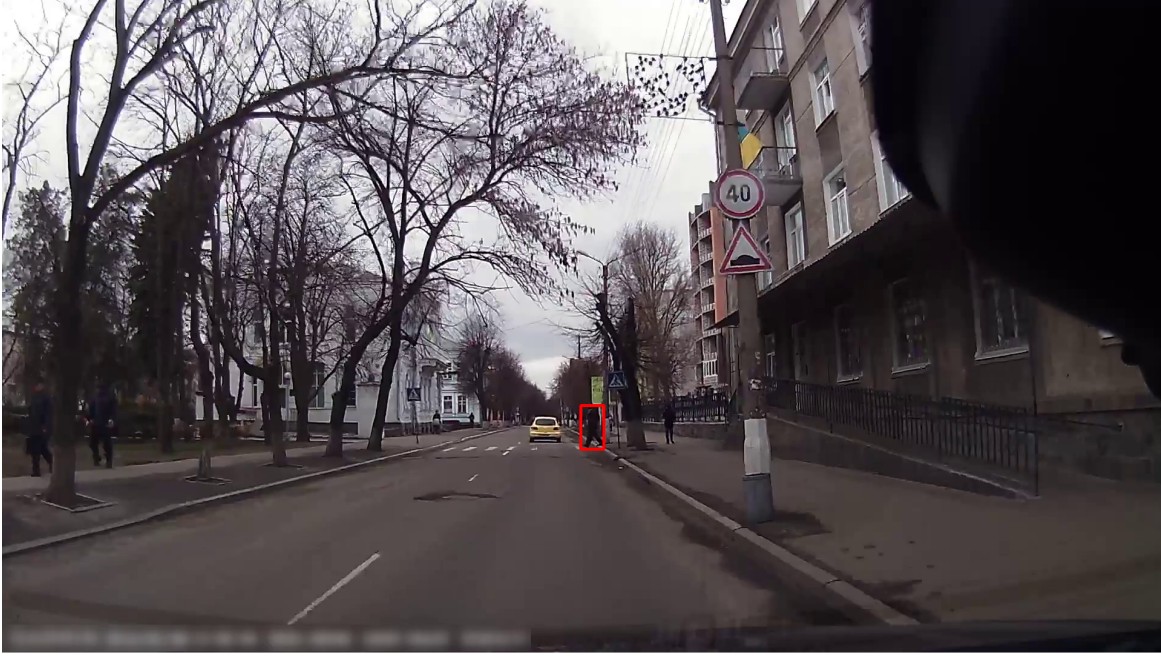

Question: Prediction Task: —Historical Bounding Box:— [[956.0, 672.0, 38.0, 72.0], [956.0, 672.0, 38.0, 72.0], [956.0, 671.0, 38.0, 73.0], [956.0, 671.0, 39.0, 73.0], [956.0, 670.0, 39.0, 74.0]]
—Pedestrian Details:— Age: senior, Gender: male, Group Size: alone —Pedestrian Behavior:— Reaction: no reaction, Hand Gesture: no, Look: not—looking, Nodding: not_nodding —Ego Vehicle Status: stopped —Environment Info:— Crosswalk: yes, Motion Direction: latitudinal, Lane Count: Two, Signalized Intersection: no, Traffic Flow: one_way, Pedestrian Crossing: yes, Pedestrian Sign: yes, Stop Sign: no, Traffic Light: no information, Road Type: street —Goal: Will the pedestrian cross the road?
Predicted Answer: Insights: — Pedestrian Info: The pedestrian is making no hand gesture and is not_nodding and not—looking toward the ego vehicle, showing a not—looking. Decision:Meta Action: THE PEDESTRIAN IS CROSSING
Actual Answer: Insights: — Pedestrian Info: The pedestrian is a male aged senior walking with a group size of alone. They are not_nodding and not—looking towards the ego vehicle, showing a no hand gesture. The ego vehicle is stopped. Decision: Meta Action: THE PEDESTRIAN IS CROSSING.

Figure 9: Examples from the JAAD dataset where the model correctly predicts "crossing" from the first frame for a given pedestrian.

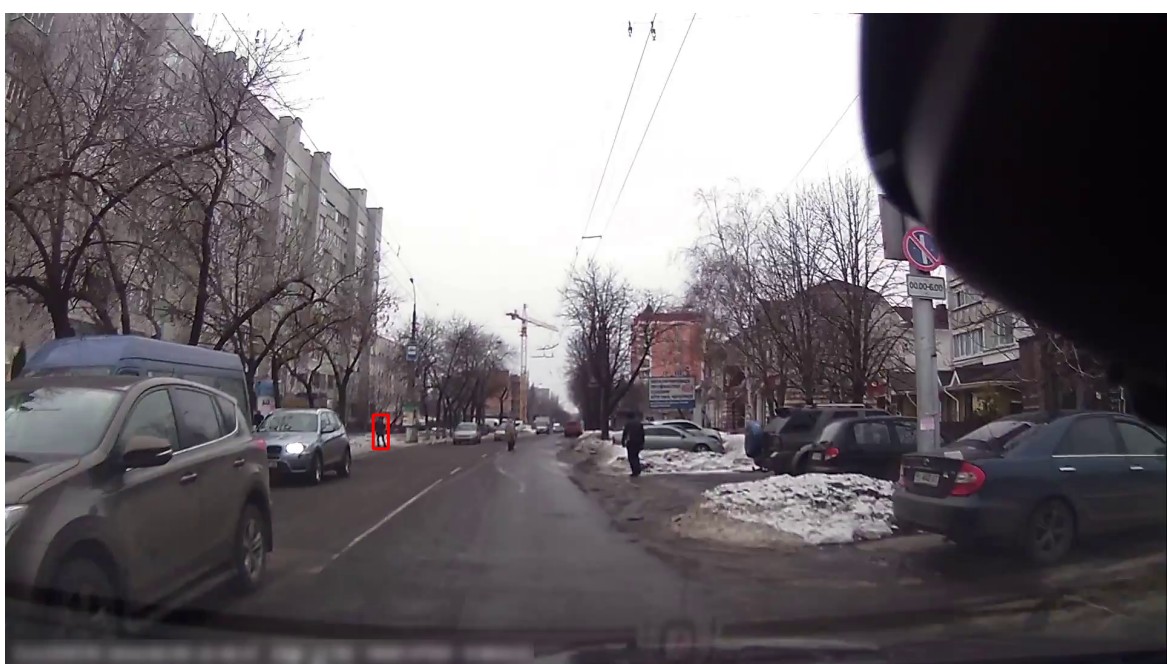

Question: Scene Analysis: —Bounding Box History:— [[608.0, 663.0, 25.0, 56.0], [609.0, 666.0, 25.0, 56.0], [610.0, 669.0, 24.0, 56.0], [611.0, 671.0, 24.0, 56.0], [612.0, 674.0, 23.0, 56.0]] —Attributes of Pedestrian:— Age: adult, Gender: female, Group Size: alone —Actions of Pedestrian:— Reaction: no reaction, Hand Gesture: no, Look: not—looking, Nodding: not_nodding —Ego Vehicle State: stopped —Scene Details: —Crosswalk: no, Motion Direction: no information, Lanes: Two, Signalized Intersection: no information, Traffic Direction: two_way, Pedestrian Crossing: no, Pedestrian Sign: no, Stop Sign: no, Traffic Light: no information, Road Type: street —Objective: Determine if the pedestrian will cross the road?
Predicted Answer: Analysis: — Pedestrian Characteristics: The pedestrian is making no hand gesture and is not_nodding and not—looking toward the ego vehicle, showing a not—looking. Conclusion:Meta Action: THE PEDESTRIAN IS NOT CROSSING.
Actual Answer: Analysis: — Pedestrian Characteristics: The pedestrian is a female aged adult walking with a group size of alone. They are not_nodding and not—looking towards the ego vehicle, with a no hand gesture. The ego vehicle is stopped.Meta Action: THE PEDESTRIAN IS NOT CROSSING.

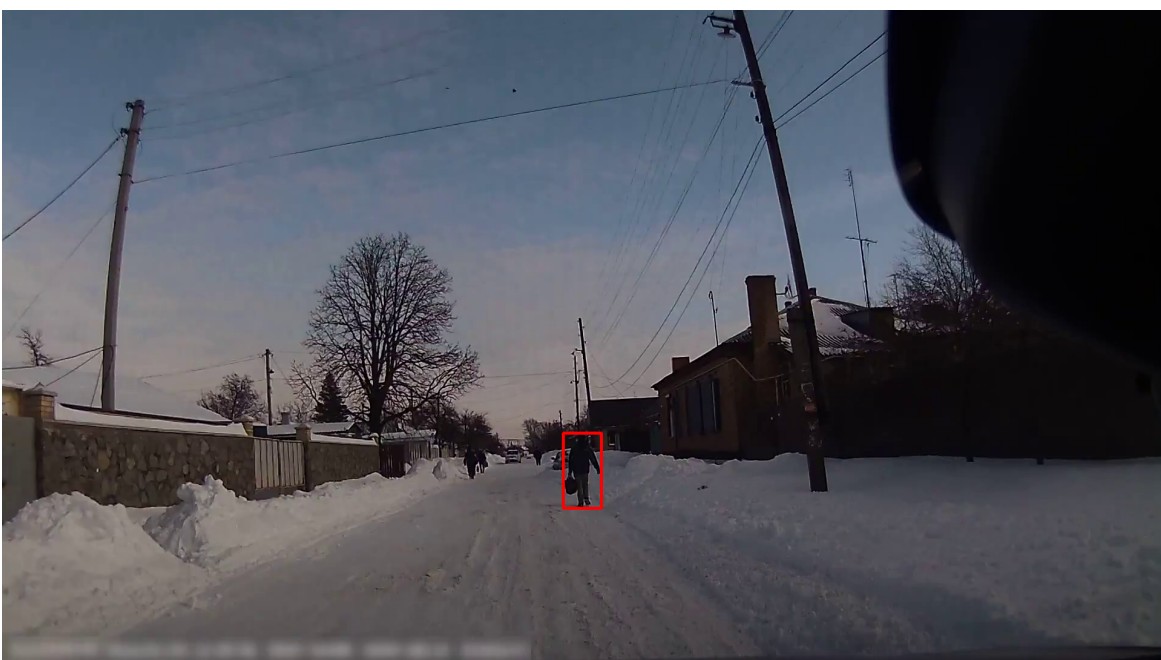

Question: Perception and Prediction: —Past History:— [[927.0, 698.0, 63.0, 124.0], [924.0, 695.0, 65.0, 126.0], [922.0, 692.0, 68.0, 127.0], [919.0, 689.0, 70.0, 129.0], [916.0, 686.0, 72.0, 130.0]] —Pedestrian Attributes:— Age: adult —Gender: male —Group Size: alone —Pedestrian Actions:— Reaction: no reaction —Hand Gesture: no —Look: not—looking —Nodding: not_nodding —The ego vehicle is stopped —Scene Information: —Designated Crosswalk: no —Motion Direction: longitudinal —Number of Lanes: Two —Signalized Intersection: no information —Traffic Direction: two_way —Pedestrian Crossing: no —Pedestrian Sign: no —Stop Sign: no —Traffic Light: no information —Road Type: street —Mission Goal: Predict if the pedestrian is going to cross the road or not?
Predicted Answer: Thoughts: — Pedestrian attribute from Perception: The pedestrian is making no hand gesture and is not_nodding and not—looking toward the ego vehicle, showing a not—looking. Meta Action: THE PEDESTRIAN IS NOT CROSSING.
Actual Answer: Thoughts: — Pedestrian attribute from Perception: The pedestrian is a male adult walking in a group of alone. The pedestrian is not_nodding and is not—looking toward the ego vehicle, showing a no hand gesture. The ego vehicle is stopped.Meta Action: THE PEDESTRIAN IS NOT CROSSING.

Figure 10: Examples from the JAAD dataset where the model correctly predicts "not crossing" from the first frame for a given pedestrian.

