# OpenReview forum: "PEDVLM: PEDESTRIAN VISION LANGUAGE MODEL FOR INTENTIONS PREDICTION"
_ICLR.cc/2025/Conference — ICLR 2025 Conference Withdrawn Submission_

### Official Review · Reviewer_buD5 · 2024-10-30

**Soundness:** 2
**Presentation:** 2
**Contribution:** 2
**Rating:** 3
**Confidence:** 4

**Summary:**

This paper addresses the lack of interpretability in existing deep learning-based methods for pedestrian intent prediction by proposing a multimodal approach that combines language models and visual modalities. This method enhances the interpretability of pedestrian intent prediction and achieves state-of-the-art performance on the JAAD dataset. Additionally, due to the absence of multimodal pedestrian intent prediction datasets, the authors have constructed the PedPrompt dataset, which integrates three previously popular pedestrian intent prediction datasets and provides new prompt annotations.

**Strengths:**

1. The authors accurately identify the current lack of interpretability in pedestrian intent prediction methods and address this issue by constructing a new dataset.
2. The provided dataset exhibits good data balance.
3. The proposed pedestrian intent prediction method effectively integrates multiple modalities, achieving excellent performance in a concise and efficient manner.
4. The authors make a clever adjustment to the loss function, resolving the model confusion caused by similar linguistic states for crossing and not crossing the road.

**Weaknesses:**

1. The authors do not provide a reasonable explanation for the significant performance drop of T5-Large+CLIP compared to T5-Base+CLIP in Table 1, which is confusing.
2. Despite the introduction of the PedPrompt dataset, there is a lack of various experiments on this dataset, including comparisons with other methods. Only statistical information about the dataset and one ablation study are provided.
3. Although the proposed method has high interpretability, its real-time performance is questionable. In practical applications, the method requires extracting optical flow and performing language model inference, both of which are time-consuming processes. The authors need to further demonstrate the feasibility of the method in real-world scenarios.
4. The authors lack comparison experiments with related methods on the TITAN dataset, which provides optical flow information. Such comparisons would better highlight the performance differences with methods that also use optical flow.
5. When explaining the poorer performance on the PIE dataset, the authors mention that PedVLM relies more on image context features but do not provide an ablation study without the text modality, making the conclusion less robust.

6.The intention prediction of this paper is limited to the scenario of crossing the road. However, in real-world settings, pedestrians may exhibit various intentions unrelated to crossing, which can still impact ego-vehicle planning, such as moving towards or away from vehicles, remaining stationary, or waiting. Expanding the range of intentions is recommended to enhance the model’s applicability and realism.

7.A comparison with other vision-language models, such as GPT-4V, is necessary to evaluate for the proposed model's performance.

8.A discussion and comparison of inference times between the proposed model and other language models should be included to provide a more comprehensive evaluation of performance.

9.The model's performance on the PIE dataset is significantly lower than other baseline models, indicating a high dependency on visual input and a lack of robustness and reasoning capabilities.

10.There are some typo errors, such as “1^-9” and “1^-4” in the implementation details, should be corrected.

**Questions:**

Please see the weakness part.

---

> ### Author Response · Authors · 2024-11-20
> **Response to Reviewer's comments**
>
> We acknowledge the reviewer for providing valuable comments and feedback. Below we provide a point-to-point response to reviewer's questions:
> 1. Larger models like T5-Large, with their higher capacity and greater number of parameters, are prone to overfitting, especially when the training data is limited (https://dataloop.ai/library/model/google-t5_t5-large/). Small datasets hinder the model's ability to learn meaningful patterns and generalize effectively, whereas a smaller model like T5-Base is less susceptible to overfitting and may perform better under these conditions. For future work, increasing the number of training example would help prevent overfitting.
> 2.  PedPrompt is developed by augmenting the TRANS dataset, which itself is constructed from three benchmark datasets widely used in the literature for pedestrian intention prediction. We have included comparisons with these benchmark datasets in Table 2 to provide context for our results. However, to the best of our knowledge, at the time of submission, there were no existing methods specifically designed for predicting pedestrian intention using vision-language models, which limits the scope for direct comparisons.
> 3. We understand the concern regarding the real-time performance of our method. In our current setup, we utilize the fast RAFT algorithm, which processes a single frame in 0.06 seconds [1]. Our T5-Base-Clip model has an inference time of 0.44 sec. Additionally, with TensorRT optimization, the T5 model inference time can be reduced to 31 milliseconds (0.031 seconds) [2]. This results in a total inference time of approximately 0.091 seconds per frame, corresponding to a processing speed of about 10.99 frames per second (10.99 Hz). Furthermore, we are exploring newer optical flow algorithms, such as NeuralFlow [3], which can achieve a frame rate of up to 30 frames per second, providing a significant improvement in processing speed. With these optimizations, our method is capable of near real-time performance, making it more feasible for practical applications. We will continue to refine and evaluate these approaches in our future work.
> 4. We would like to clarify that we used optical flow for all the datasets, not just TITAN. To the best of our knowledge, at the time of submission, there is no published work that has used the TITAN dataset for pedestrian intention prediction, as the TITAN dataset is more general in nature. We included the TITAN dataset in our study primarily to enhance the diversity and increase the number of training examples for our dataset. Additionally, to the best of our knowledge, there is no existing algorithm that uses optical flow specifically for predicting pedestrian intention.
> 5. We acknowledge the concern regarding the explanation of our model's performance on the PIE dataset. The poorer performance is primarily observed on pedestrians that are very far from the camera, where image context features alone are less effective due to reduced visual detail. Regarding the reviewer's suggestion to conduct an ablation study without the text modality, we believe that removing text input would not be appropriate in this context. Our model is designed to query specific pedestrian intentions, and relying solely on image features would limit its ability to make accurate predictions. Moreover, removing the text modality would fundamentally change the nature of the model, as it would no longer function as a vision-language model.
> 6. The pedestrian intention problem is well established in the literature [4,5], where the pedestrian behaviors are being classified as "crossing" and "not crossing". However, we acknowledge, other behaviors like moving towards the ego-vehicle, stationary, etc, exists, but these behaviors can be regarded as sub-behaviors that can be considered under the umbrella of "crossing" and "not crossing" behaviors. Additionally, manually labeling such diverse behaviors is a labor-intensive task which we left as a future work based on our current work.
>
> **References**
> 1. A. Zuzow and C. Nimo, “Real-time optical flow estimation, https://charlesnimo.me/files/comp_vision.pdf"
> 2. V. Nguyen, N. Srihari, P. Chadha, C. Chen, J. Lee, and J. Rodge, “Optimizing t5 and gpt-2 for
> real-time inference with nvidia tensorrt (2021).”
> 3. Z. Zhang, H. Jiang, and H. Singh, “Neuflow: Real-time, high-accuracy optical flow estimation on
> robots using edge devices,” arXiv preprint arXiv:2403.10425, 2024
> 4. C. Zhang and C. Berger, “Pedestrian behavior prediction using deep learning methods for urban scenarios: A review,” IEEE Transactions on Intelligent Transportation Systems, vol. 24, no. 10,pp. 10279–10301, 2023.
> 5. A. Rasouli, I. Kotseruba, and J. K. Tsotsos, “Are they going to cross? a benchmark dataset and baseline for pedestrian crosswalk behavior,” in Proceedings of the IEEE International Conference on Computer Vision Workshops, pp. 206–213, 2017.

---

> ### Author Response · Authors · 2024-11-20
> **Response to Reviewer's comments**
>
> 7. We have presented the comparison of our PedVLM on JAAD dataset with GPT-4V model. However we reported the GPT-4V results as illustrated here [1] in our comparison. We acknowledge that evaluating our method with a different large language model such as GPT-4V would produce more insights, but in this work we have focused our directions toward using open-source version of large language model such as T5 for the framework development. Additionally, due to resource constraints, we restrict ourself to use the open versions of large-language models.
>
> 8. We appreciate the suggestion to include a discussion and comparison of inference times to provide a more comprehensive evaluation. Our proposed model has an inference speed of approximately 2.27 frames per second (0.44 sec) without any optimizations. However, there is no specific vision-language model (VLM) that uses the exact same configuration as our method, making direct comparisons challenging and potentially unfair.
>
> 9. We acknowledge that the model's performance on the PIE dataset is lower compared to other baseline models, highlighting a dependency on visual input and suggesting areas where robustness and reasoning capabilities could be improved. However, as part of our ongoing research, understanding and addressing these limitations is a key focus, and we view this as an opportunity for future work. Moreover, our findings emphasize the crucial role of vision in vision-language models, analogous to the human brain, where visual input is fundamental for understanding and predicting the environment. Just as humans rely on what they can see to make informed decisions, our model demonstrates a similar reliance. We remain optimistic about our results, as they underscore the potential for further development and refinement to enhance performance and reasoning capabilities. We will address this discussion in the revised manuscript to clarify our research perspective and the path forward.
>
> 10. The typos are fixed, thank you for mentioning them.
>
> **References**
> 1. J. Huang, P. Jiang, A. Gautam, and S. Saripalli, “Gpt-4v takes the wheel: Evaluating promise and challenges for pedestrian behavior prediction,” arXiv preprint arXiv:2311.14786, 2023.

---

> ### Author Response · Authors · 2024-11-28
>
> Thank you for your response and for taking the time to review our work. While we respect your decision to maintain your final rating, we believe that simply stating that the rebuttal does not address your concerns, without specifying those concerns in detail, does not provide a constructive path for improving the research. Clear and specific feedback is essential for meaningful revisions, and we would appreciate further elaboration on your points to better address your concerns.

---

### Official Review · Reviewer_mQpi · 2024-10-30

**Soundness:** 2
**Presentation:** 2
**Contribution:** 2
**Rating:** 3
**Confidence:** 4

**Summary:**

The authors argue that traditional deep learning methods fall short of capturing the complexity of pedestrian behavior, often lacking in explainability in the context of autonomous vehicles. To address this problem, they introduce PedVLM, a novel vision-language model designed to predict pedestrian intentions and provide interpretable explanations for those predictions in self-driving. This model integrates multiple modalities, including RGB images, optical flow, and text, to enhance the model's contextual understanding and predictive accuracy. Along with this model, the PedPrompt dataset is introduced based on existing datasets like JAAD, PIE, and TITAN, which consists of Question-Answer (QA) prompts for pedestrian intention prediction. PedVLM is evaluated on the PedPrompt dataset, which surpasses GPT-4V and other traditional methods in terms of F1-score and AUC on JAAD.

**Strengths:**

* This research proposes an effective VLM model, which integrates RGB for temporal understanding and optical flow for temporal perception.
* This work formulates one pedestrian intention prediction dataset, which includes 48,696 prompts. This is a significant contribution to the field of pedestrian intention prediction.

**Weaknesses:**

* **The lack of related work discussion**

     The discussion on VLMs for driving environment understanding is not enough. Especially recently, there have been lots of related works using VLMs to provide interpretable explanations for traffic conditions and decisions. Some representative examples are listed below. Including them and other relevant references will strengthen the contribution of the work.

      [1]Sima C, Renz K, Chitta K, et al. Drivelm: Driving with graph visual question answering. ECCV 2024.
      [2]Marcu A M, Chen L, Hünermann J, et al. Lingoqa: Video question answering for autonomous driving[J]. arXiv preprint arXiv:2312.14115, 2023.
      [3]Malla S, Choi C, Dwivedi I, et al. Drama: Joint risk localization and captioning in driving. WACV 2023.
      [4]Chen L, Sinavski O, Hünermann J, et al. Driving with llms: Fusing object-level vector modality for explainable autonomous driving[C]//2024 IEEE International Conference on Robotics and Automation (ICRA). IEEE, 2024.
      [5]Han W, Guo D, Xu C Z, et al. Dme-driver: Integrating human decision logic and 3d scene perception in autonomous driving[J]. arXiv preprint arXiv:2401.03641, 2024.

---

* **The limitation of the proposed PedPrompt**

   1. This dataset is specifically curated to focus on pedestrian intention prediction. However, real-life driving scenes include various conditions, like road layout, dynamic vehicles, and other obstacles [1]. The explanations provided by this dataset are very limited.
   2. PedPrompt only provides interpretable explanations for a single object within a video sequence. Regrettably, this object may not necessarily be a critical influence on driving decisions, potentially resulting in misleading driving attention and subsequent inaccuracies in predictive models.
   3. The explanation answer only includes one kind of action for pedestrians, that is, “crossing” or “not “crossing”. This simplistic categorization fails to capture the complexity of pedestrian behavior. For instance, a pedestrian may indeed be crossing the road but not directly impacting the ego vehicle's trajectory. Thus, a more detailed description to understand and predict intention could be more helpful for this benchmark.
    4. In this dataset, various prevalent driving datasets, such as BDD and nuScenes, have not been taken into account.

---
Overall, this work using VLM for pedestrian intention prediction misses an important literature review and provides a dataset with several key limitations.

**Questions:**

* How does the length of the input frame sequence affect the model's ultimate performance? Is the 5-frame input enough for pedestrian intention prediction?

---

> ### Author Response · Authors · 2024-11-20
> **Response to Reviewer's comments**
>
> We appreciate the reviewer for positive feedback.
>
> **Lack of related work discussion**: We have focused the discussion of related work mostly on pedestrian intent prediction.
> But we agree that use of VLMs for other tasks and general understanding of the driving situation can be beneficial for setting the context of our work, as the approaches for environment understanding could be beneficial also for the concrete task of pedestrian intent prediction. Therefore, we will extend the related work section to cover also other aspects of use of foundation models for understanding driving situations, including the mentioned works.
>
> **Limitations of the proposed PedPrompt**:
>
> We are not aiming towards predicting a comprehensive description of the driving situation and all its components, but following previous work [1,2], we focus on the task of pedestrian intent prediction, where the objective is to predict whether a given pedestrian is crossing the road or not. While this indeed does not give a full understanding of the scene, it aims at solving one component of it. The existing datasets we base our dataset on also focus on this specific task, and they have been annotated so that they give information about specific pedestrians and information about whether they are crossing or not. We follow the same setup. Popular general driving datasets, such as the suggested BDD and nuScenes, do not explicitly contain information per pedestrian. We will mention these datasets in the overview of explainability from the previous suggestion.
>
>
> **Question: How does the length of the input frame sequence affect the model’s ultimate performance? Is the 5-frame input enough for pedestrian intention prediction?**
> Thank you, this is a very good question. It is true that the number of frames affects the prediction accuracy of the models. In previous works, usually more frames are used - for example, in the work we mention in Table 2, they use 10, 16 or 32 frames.
> Using fewer frames and obtaining similar results to previous works, shows the strength of using VLMs for pedestrian prediction.
> We will explain this in the paper. Indeed, conducting results with longer history of more frames is a good idea and this is definitely something that we will explore in future work.
>
> **References**
> 1. C. Zhang and C. Berger, “Pedestrian behavior prediction using deep learning methods for urban scenarios: A review,” IEEE Transactions on Intelligent Transportation Systems, vol. 24, no. 10,pp. 10279–10301, 2023.
> 2. A. Rasouli, I. Kotseruba, and J. K. Tsotsos, “Are they going to cross? a benchmark dataset and baseline for pedestrian crosswalk behavior,” in Proceedings of the IEEE International Conference on Computer Vision Workshops, pp. 206–213, 2017.

---

> ### Author Response · Authors · 2024-11-24
> **Response to Reviewer Comment**
>
> We appreciate the reviewer’s perspective; however, we hold a differing viewpoint on this comment. Autonomous driving encompasses several key components, including perception, planning, prediction, and control. Within the perception domain of autonomous vehicles (AVs), various tasks such as segmentation, object detection, and lane detection are addressed. In our work, we clearly stated that the scope of our research is focused on learning pedestrian behaviors, excluding the behavior of other vehicles, which falls under the motion planning paradigm. Our objective is to predict pedestrian behavior, primarily due to the challenges posed by the intricate and unpredictable nature of human motion patterns.
>
> In the literature, pedestrian intention prediction is primarily treated as a distinct classification problem, typically focused on crossing vs. non-crossing actions. While datasets like nuScenes are valuable for broader perception tasks, including motion planning, prediction, and detection, they lack pedestrian-specific attributes and do not provide detailed labels for pedestrian decision-making.
> Therefore, benchmark datasets such as JAAD and PIE, which are widely used in this domain and are well-established in the research community as standard benchmarks for studying pedestrian behavior, were employed in our study to address this focused challenge. Our objective was to explore the potential of vision-language models (VLMs) in predicting pedestrian behavior. The proposed method shows promising results, providing a foundation for further research and highlighting directions for future work in this domain. We appreciate your insights and will consider them to refine and strengthen our contributions.

---

### Official Review · Reviewer_EL9a · 2024-11-03

**Soundness:** 2
**Presentation:** 3
**Contribution:** 1
**Rating:** 3
**Confidence:** 4

**Summary:**

This paper proposes an augmented dataset denoted as PedPromt, which is based on existing TRANS dataset. Through mannually designed prompt template, this paper introduce a framework to predict pedestrian behavior and generate related explainations simultaneously. Compared with existing state-of-the-art VLM, the proposed model achieves better performanceon PIE and JAAD dataset.

**Strengths:**

1. This paper introduces augmented language annotations on existing TRANS dataset and make it suitable to generate explanations of pedestrian behavior.

2. The proposed model achieves better performance than state-of-the-art large VLMs like GPT-4V on PIE and JAAD dataset

**Weaknesses:**

1. Writing issues:
 (a)  All figures are too blurry to be seen clearly.
(b) In Section 4.2, the index 𝑖 of visual embeddings should not be used to represent the projection matrix 𝑃.

2. The contribution to the TRANS dataset is insufficient. While VLMs are a popular topic, many existing works augment datasets with language explanations. This paper’s template incorporates substantial information from the TRANS dataset but should further explore ways to expand upon this foundation.

3. The exploration of the prompt template is inadequate and does not ensure that the proposed template is optimal. Current LLMs are not sensitive to numerical data, which could lead to issues when numbers are directly translated to strings as input.

4. The contribution of the proposed method is limited. The paper mainly focuses on merging different modalities using simple modules. The authors should place greater emphasis on ensuring effective feature alignment between modalities.

**Questions:**

The quality of the augmented dataset should be addressed. The hallucination from LLM should be discussed.

---

> ### Author Response · Authors · 2024-11-20
> **Response to Reviewer's comments**
>
> We acknowledge the reviewer for the positive feedback. Below we provide point-to-point response to the reviewers comments:
> 1. We have updated the manuscript figures and resolved other formatting issues.
> 2. The TRANS dataset, which we use in our work, is a combination of three widely adopted real-world pedestrian intention prediction datasets: JAAD, PIE, and TITAN. These datasets are frequently used in the literature for pedestrian intention prediction tasks as they provide challenging scenarios and complex pedestrian behaviors. The TRANS dataset augments these datasets by providing detailed annotations on pedestrian actions (e.g., "walk," "stand," "stop," "go") and transitions between these states, making it highly suitable for our task. Given the significant effort required for manual data collection and labeling, we opted to use the TRANS dataset, which is already available and well-suited for pedestrian intention prediction. We further augmented it with language prompts to create the PedPrompt dataset, which enhances the explainability of pedestrian behavior by framing the problem in a question-answer format. This augmentation allows us to leverage vision-language models (VLMs) effectively while maintaining consistency with existing benchmarks in the field. Additionally, our evaluation shows that PedVLM outperforms state-of-the-art methods on standard pedestrian intention prediction metrics across these challenging datasets.
> 3. We acknowledge the reviewer's concern about the sensitivity of large language models (LLMs) to numerical data. Our work addresses this issue by converting numerical data, such as bounding box information, into text strings. This conversion allows the T5 model to process the data effectively using its tokenizer. This approach is well-established in the literature, where bounding box or trajectory information is often transformed into text for use with LLMs [1,2]. Regarding the design of prompt templates, we have implemented various design patterns to capture a wide range of prompts. Our experimental results demonstrate that our proposed prompt design enables PedVLM to outperform state-of-the-art methods in pedestrian intention prediction accuracy and explainability metrics.
> 4. PedVLM's novelty lies in its application-specific framework for pedestrian intention prediction, which goes beyond simply merging modalities. PedVLM integrates multiple sensory inputs (RGB images, optical flow, and text) to create a contextual representation of the environment. This integration is achieved through a gated-attention pooling mechanism, which ensures effective feature alignment between the visual and textual modalities, allowing the model to dynamically weigh the importance of each modality based on the context. Moreover, PedVLM introduces a novel PedPrompt dataset, which augments the TRANS dataset (a combination of JAAD, PIE, and TITAN) with language prompts explicitly designed for pedestrian intention prediction. This dataset enhances explainability by framing the problem in a question-answer format, enabling the model to generate interpretable explanations alongside accurate predictions. Our experimental results demonstrate that PedVLM outperforms state-of-the-art methods in pedestrian intention prediction accuracy and explainability metrics. This indicates that our approach to feature alignment and multimodal integration effectively handles complex real-world scenarios.
> 5. In our work, we conducted an extensive experimental evaluation of the PedPrompt dataset using baseline methods. Additionally, we assessed PedVLM's quantitative and qualitative performance to identify failure cases where hallucinations occur. These hallucinations involve the model generating explanations that do not accurately reflect the actual scene information. To illustrate these discrepancies, we have included examples of such failure cases in the supplementary material of our paper. Despite these occasional hallucinations, our experimental results show that PedVLM consistently surpasses state-of-the-art methods in both pedestrian intention prediction accuracy and explainability metrics. However, we recognize that addressing hallucinations remains a persistent challenge with large language models. Further refinement of prompt design and model training could help mitigate this issue in future work.
>
> **References**
> 1. Mao, J., Qian, Y., Ye, J., Zhao, H. and Wang, Y., 2023. Gpt-driver: Learning to drive with gpt. arXiv preprint arXiv:2310.01415.
> 2. Chen, Long, Oleg Sinavski, Jan Hünermann, Alice Karnsund, Andrew James Willmott, Danny Birch, Daniel Maund, and Jamie Shotton. "Driving with llms: Fusing object-level vector modality for explainable autonomous driving." In 2024 IEEE International Conference on Robotics and Automation (ICRA), pp. 14093-14100. IEEE, 2024.

---

### Official Review · Reviewer_XLVi · 2024-11-04

**Soundness:** 2
**Presentation:** 1
**Contribution:** 1
**Rating:** 3
**Confidence:** 3

**Summary:**

The paper presents PedVLM, a novel Vision-Language Model (VLM) designed to predict pedestrian intentions in urban environments and improve the safety of autonomous vehicles. Traditional deep learning methods have struggled with accurately capturing the complexities of pedestrian behavior and often lack explainability. To address this, PedVLM combines visual data (RGB images and optical flow) with textual descriptions using a CLIP-based vision encoder and a T5 language model. This integration allows the model to predict pedestrian actions, such as crossing or not crossing while providing interpretable explanations for its decisions.
A key contribution of the work is the development of the PedPrompt dataset, which includes question-answer style prompts tailored for pedestrian intention prediction. The authors evaluate PedVLM on various datasets, including PedPrompt, JAAD, and PIE, showing that it outperforms state-of-the-art models, particularly in terms of F1-score and AUC for pedestrian intention prediction. However, the study also notes performance challenges with pedestrians who are distant or partially occluded, highlighting areas for future improvement.

**Strengths:**

- The paper introduces a comprehensive dataset, PedPrompt, which supports further research in pedestrian intention prediction. This dataset is meticulously curated, including a variety of contextual and pedestrian behavior information
- Unlike many deep learning models that act as black boxes, PedVLM emphasizes explainability. By using language models to provide reasoning for predictions, the model enhances interpretability, which is crucial for autonomous driving applications.

**Weaknesses:**

- The model's performance significantly depends on the visibility and clarity of pedestrian cues. As noted in the evaluation, PedVLM struggles with distant or partially occluded pedestrians, particularly in the PIE dataset, which limits its robustness in complex urban environments.
- The model simplifies pedestrian intention to crossing versus non-crossing (Simplified Binary Classification), which may not capture the full complexity of pedestrian behavior, such as hesitation or erratic movements, limiting its applicability in real-world situations.
- The paper does not convincingly justify the necessity of using Vision-Language Models (VLMs) for pedestrian intention prediction. The explanations generated by PedVLM, such as “The pedestrian is making no hand gesture, is not nodding, and is looking toward the ego vehicle,” seem to reiterate the attributes explicitly mentioned in the input prompts. This raises concerns about whether the model is genuinely interpreting and understanding visual cues from the images or simply parroting back the information provided in textual form, rather than making independent inferences based on visual and contextual analysis.

**Questions:**

- Since pedestrian intention prediction here is a simple binary task (crossing or not crossing), why use a complex language model like T5? Wouldn’t a simpler model work just as well, or is there a specific benefit to using T5 that makes it worth the added complexity?
- The explanations generated by PedVLM often seem to mirror the attributes provided in the prompts, raising concerns about the model’s interpretive abilities. How can you demonstrate that the model is not just repeating input information but is genuinely understanding and interpreting visual cues?
- How would PedVLM handle real-world conditions, such as complex pedestrian behaviors or environments with limited visibility and frequent occlusions? Have you considered evaluating the model under more challenging or diverse conditions?

---

> ### Author Response · Authors · 2024-11-20
> **Response to Reviewer's comments**
>
> We appreciate the reviewer for valuable feedback and acknowledge the limitation of PedVLM's strong dependence on visual cues for predicting the intentions for very far pedestrians. Our current method relies solely on camera images without incorporating other sensor modalities such as LiDAR or radar, which offer greater robustness and spatial resolution. To mitigate this limitation, we have implemented image-based enhancements, such as optical flow analysis to capture motion and complement static image data in observing pedestrian intentions. However, we recognize that integrating PedVLM with additional sensor modalities is a crucial area for future work. This integration will address the current limitations of our approach and potentially lead to a more comprehensive and reliable system for pedestrian intention prediction across various environmental conditions.
>
> Regarding the simplification of pedestrian intention to a binary classification problem, we believe that focusing on crossing and not-crossing behaviors aligns closely with the primary safety-critical decisions required in road environments. While other behaviors, such as erratic movement and hesitation, are important, we consider these to be sub-behaviors that can be explained within the context of crossing and not-crossing intentions. Furthermore, it is well-established in the literature that pedestrian intention is typically observed and modeled as crossing and not-crossing behaviors [1,2].
>
> In addition, the close correlation between input prompts and generated explanations is expected due to the structure of our task, where both inputs and outputs are aligned to reduce loss during training. This ensures that the model produces consistent outputs that match the input format. However, PedVLM is not merely repeating information; it combines visual embeddings from RGB images and optical flow with text embeddings to generate predictions based on multimodal data. The model interprets visual cues (e.g., gestures, gaze direction) and contextual information to make informed predictions. Additionally, we evaluate PedVLM's performance both qualitatively and quantitatively to ensure that it genuinely learns from visual data and not simply parrots back input prompts. The quantitative results (e.g., F1-score, AUC) demonstrate that PedVLM achieves competitive performance in intention prediction tasks, while qualitative assessments show that its explanations align with observed pedestrian behaviors.
>
> **References**
> 1. C. Zhang and C. Berger, “Pedestrian behavior prediction using deep learning methods for urban scenarios: A review,” IEEE Transactions on Intelligent Transportation Systems, vol. 24, no. 10,pp. 10279–10301, 2023.
> 2. A. Rasouli, I. Kotseruba, and J. K. Tsotsos, “Are they going to cross? a benchmark dataset and baseline for pedestrian crosswalk behavior,” in Proceedings of the IEEE International Conference on Computer Vision Workshops, pp. 206–213, 2017.

---

> ### Author Response · Authors · 2024-11-20
> **Response to Reviewer's questions**
>
> Below, we give point-to-point responses to reviewer's questions:
> 1. The choice of T5 is driven by its ability to handle complex multimodal inputs and generate high-quality, context-aware explanations, which are critical for our task. While pedestrian intention prediction is binary, the challenge lies in integrating diverse visual and contextual information (e.g., pedestrian actions and scene dynamics) to make accurate predictions and provide interpretable explanations. T5 excels in processing text-based prompts and generating meaningful, context-rich outputs that simpler models may struggle to achieve. Furthermore, we have evaluated the performance of PedVLM against other state-of-the-art methods, including simpler GRU-based models. Our results demonstrate that PedVLM consistently outperforms these methods in predictive accuracy and explainability metrics, showcasing its efficacy despite the added complexity. This indicates that the use of T5 is justified by its contribution to prediction performance and the quality of generated explanations.
>
>
> 2. We acknowledge the concern regarding the correlation between input prompts and generated explanations. However, PedVLM is designed to integrate multimodal inputs (RGB images, optical flow, and text) to interpret and reason about pedestrian behavior genuinely. The close alignment between input prompts and generated explanations results from our loss function, which requires the model to output predictions in a format that minimizes loss by maintaining consistency with the input structure. This ensures the model learns to map visual cues (e.g., gestures, gaze direction) to specific textual descriptions. To demonstrate that PedVLM is not merely repeating input information but is genuinely interpreting visual cues, we evaluate its quantitative and qualitative performance. Quantitatively, PedVLM shows competitive results on standard pedestrian intention prediction metrics (e.g., F1-score, AUC) across multiple datasets (JAAD, PIE) compared to state-of-the-art methods. Qualitatively, we assess the generated explanations alongside visual inputs to ensure that they reflect meaningful interpretations of pedestrian behavior. For example, when a pedestrian's gaze direction or body posture indicates hesitation, PedVLM generates corresponding explanations that are contextually aligned with these visual cues rather than simply restating input attributes. Additionally, we also showed that in our supplementary material, the failure cases were where the PedVLM produced wrong or misinterpreted explanations for the pedestrian intention prediction.
>
>
> 3. PedVLM has been evaluated on the JAAD, and PIE datasets collected from real-world driving scenarios. These datasets include challenging conditions such as complex pedestrian behaviors and varying environmental factors. To ensure a fair comparison with state-of-the-art methods, we adopted standard evaluation metrics used by the research community. Our results demonstrate that PedVLM outperforms other state-of-the-art methods, showcasing better efficacy in both pedestrian intention prediction and explainability.

---

> ### Author Response · Authors · 2024-11-24
> **Response to reviwer Comments**
>
> In our work, we explore the application of vision-language models (VLMs) specifically for pedestrian intention prediction. The model performs well on some datasets and provides comparable results on others, as detailed in our paper. We have also discussed its limitations, which highlight areas for future research and improvements needed for real-world applicability. Despite these challenges, the significant improvements achieved using VLMs demonstrate their potential for advancing this domain. Having
> said that, answer your individual concerns.
>
> Indeed, the models that have shown better performance on the PIE dataset utilized camera images augmented with pedestrian keypoint information specifically designed for pedestrian intention prediction. Our method further extends this approach and makes it challenging by incorporating explanability for pedestrian intention behavior. We use camera images and optical flow images augmented with text to predict and explain pedestrian intentions. Furthermore, combining camera data with other sensor modalities, such as radar and LiDAR, could potentially improve PedVLM results. This is one of the approaches we mentioned, but it’s important to clarify that we did not state that LiDAR and radar must be incorporated with camera data to enhance PedVLM performance. PedVLM performance based solely on images can be further improved by incorporating more specific local contextual features and utilizing more advanced spatio-temporal VLM-based models. This approach offers an alternative path for enhancement without necessarily relying on additional sensor modalities. Addressing the concern about real-world applications, we concur that practical implementation involves numerous factors beyond the scope of our current paper. One such consideration is safety, which is required for both pedestrians and ego-vehicles. While redundant sensor modalities may increase costs, they shouldn’t be dismissed when safety is at stake. Our research lays the groundwork for future studies that could explore how to integrate multiple sensors to enhance real-world applicability and better models for learning pedestrian behavior. Regarding the effectiveness of our Large Language Model(LLM), we acknowledge the reviewer’s viewpoint, but we have a different point of view on that. Our extensive experimental results demonstrate the LLM’s significant contribution to understanding and predicting pedestrian behavior, particularly in the challenging scenarios present in the JAAD dataset. In this work, we employ a question-and-answer (QA) format and design prompt templates in various Q&A styles to add diversity to our prompts. Our specific intention is for the answers to align with the question format, providing a clear understanding of what the scene represents and offering insights
> into pedestrian behaviors. By utilizing this format, we have specifically designed the loss function to address both explanability and pedestrian intention prediction. Furthermore, our linguistic evaluation metrics (e.g., BLEU-4, METEOR) confirm the quality of the generated explanations, reinforcing that the LLM is not merely reiterating input prompts but actively reasoning based on multimodal inputs. To illustrate this point, we have included examples of failure cases in our supplementary material, demonstrating that our PedVLM does not simply parrot the questions. These aspects collectively support our assertion that the LLM plays a crucial role in our system’s performance and contributes meaningfully to the task at hand.

---

### Author Response · Authors · 2024-11-20
**Brief Summary**

We sincerely thank the ICLR reviewing committee and the reviewers for their valuable and constructive feedback on our work.
We appreciate that the reviewers acknowledge the **strengths** of our work:
1. The effectiveness of the PedPrompt dataset relies because of its balanced data, multiple modalities and variety of contextual and pedestrian behavior information.

2. The good performance of the PedVLM model and its contribution to increasing explainability in pedestrian intent prediction.

3. The originality of the adjustment to the loss function, resolving the model confusion caused by similar linguistic states for crossing and not crossing the road.


Below, we provide a **brief context and summary of the key comments**, which we believe will assist in the further review of our paper. Additionally, this summary aims to offer readers a clearer understanding of our contributions and how we have addressed the concerns raised by the reviewers.

1. PedVLM's novelty relies in designing an application-specific framework for pedestrian intention prediction using vision-language model (VLMs).

2. The key contribution of our work is the creating of PedPrompt dataset, developed by augmenting the TRANS dataset with language prompts, enhancing both prediction accuracy and explainability. The TRANS dataset combined three benchmark data for pedestrian intent prediction (JAAD, PIE and TITAN). This integration allows PedPrompt to serve as a more diverse and comprehensive dataset, enabling models to generalize better across different scenarios and environments.

3. Furthermore, in our work we classified the pedestrian behavior into "crossing" and "not crossing", which are being well established in literature as nomenclature to address the pedestrian intention prediction [1,2]. Other pedestrian behaviors also exist, but most of them can be considered as sub-behaviors under the umbrella of "crossing" and "not crossing".

4. The efficacy of the PedVLM is experimentally evaluated on PedPrompt dataset by designing the baseline methods and also on popular benchmark datasets (JAAD and PIE) to make a fair comparison with state-of-the-art methods. Through our experiments, PedVLM illustrates better performance in challenging scenarios occurred in JAAD dataset, whereas comparable results in PIE dataset. Since, to the best of our knowledge there were no available work that have used TITAN for pedestrian intention at the time of submission, so we did not evaluate the PedVLM performance on TITAN dataset.

5. Since our PedVLM framework is heavily dependent on the visual cues, the caveat of this one involves that the pedestrian which may appear very far in the image space or get occluded, PedVLM struggles to generate the explanation for that particular behavior, which is how human behaves. In our opinion the performance could be improved using different sensors, like radar and LiDAR.

6. Deploying in real-world applications is an issue of future work, as with most of the recent works on large (visual-)language models. We believe that PedVLM could be used in real-world applications by designing proper optimizations for embedded hardware. For example, it has been shown that T5 model inference with TensorRT optimization yield to 0.031 seconds [3] (for comparison, in our setup T5-Base-Clip inference is 0.44 seconds). In the current settings, we have used RAFT [4] for optical flow that processes a single frame in 0.06 seconds.

7. Another plausible concern that reader may feel is the use of other large-language model, for instance GPT-4V to our problem settings. Our model outperforms the results from recent work that uses GPT-4V [5] in a zero-shot setting. It is possible to fine-tune also GPT-4V for the specific task, but in this work we have focused our research direction to use open-source version of large-language models for instance T5 model.


**We will update the manuscript shortly**

**References**
1. C. Zhang and C. Berger, “Pedestrian behavior prediction using deep learning methods for urban scenarios: A review,” IEEE Transactions on Intelligent Transportation Systems, vol. 24, no. 10,pp. 10279–10301, 2023.
2. A. Rasouli, I. Kotseruba, and J. K. Tsotsos, “Are they going to cross? a benchmark dataset and baseline for pedestrian crosswalk behavior,” in Proceedings of the IEEE International Conference on Computer Vision Workshops, pp. 206–213, 2017.
3. V. Nguyen, N. Srihari, P. Chadha, C. Chen, J. Lee, and J. Rodge, “Optimizing t5 and gpt-2 for real-time inference with nvidia tensorrt (2021).”
4. A. Zuzow and C. Nimo, “Real-time optical flow estimation, https://charlesnimo.me/files/comp_vision.pdf"
5.  J. Huang, P. Jiang, A. Gautam, and S. Saripalli, “Gpt-4v takes the wheel: Evaluating promise and challenges for pedestrian behavior prediction,” arXiv preprint arXiv:2311.14786, 2023.

---

### Note · Authors · 2024-11-29

I have read and agree with the venue's withdrawal policy on behalf of myself and my co-authors.